# Wearable devices for anxiety assessment: a systematic review

Mohamed Elgendi [1,2,8] ✉, Karmen Markov[3,8], Hangyu Liu[3,4], Maarten De Vos[5], Kinda Khalaf[1],
Ahsan Khandoker[1], Herbert F. Jelinek [6], Debkalpa Goswami [3,7] & Carlo Menon [3] ✉

## Abstract

**Background** There is growing interest in using biosignals from wearable devices to assess anxiety disorders. Among these, electrocardiography is the most widely used due to its ability to monitor cardiovascular activity. Other signals, such as respiratory, electrodermal activity, and photoplethysmography, also show promise. This review aims to evaluate how these signals, individually and in combination, have been used for anxiety detection.
**Methods** We systematically reviewed 26 studies published between 2014 and 2024 that used wearable devices to collect signals for anxiety detection. Extracted information included study design, signal types, features, classification methods, and accuracy outcomes. Pooled accuracies were calculated to compare single-signal and multi-signal approaches.
**Results** Here we show that approaches combining multiple signals outperform those using a single signal, with a pooled accuracy of 81.94% compared to 76.85%. Electrocardiography was the most reliable individual signal, with a pooled accuracy of 80.34% across 12 studies. However, the limited number of single-sensor studies and methodological variability limit conclusions about the superiority of any one modality. The most common features included mean heart rate and heart rate variability for electrocardiography, the mean inspiratory-to-expiratory time ratio for respiratory signals, mean skin conductance for electrodermal activity, and the mean heart rate for photoplethysmography. Support vector machine was the predominant classifier.
**Conclusions** This review underscores the clinical potential of wearable devices for anxiety detection, emphasizing the value of multimodal approaches. Future research should focus on refining algorithms, expanding sample sizes, and exploring diverse contexts to improve the accuracy and generalizability of these methods.

## Plain Language Summary

Anxiety disorders are common, but detecting them in everyday life can be difficult. Wearable devices, such as smartwatches, can record body signals that change with anxiety, including heart activity, breathing, skin responses, and blood flow. We reviewed 26 studies from the past decade that tested these signals for detecting anxiety. We found that combining several signals gives more accurate results than using a single signal, while heart activity was the most reliable single signal. However, most studies were small, used different devices and methods to trigger and measure anxiety, making it difficult to draw firm conclusions. Overall, wearable devices show strong potential for anxiety detection, but future research needs larger, standardized studies before they can be widely used in daily life and healthcare.

According to the World Health Organization, anxiety disorders (ADs) are the world's most common mental health challenge, affecting over 3.6% of the world's population, with worse impact in developing countries[1]. It is also estimated that, globally, one in seven (14%) of 10-19-year-olds experience mental health disorders, with depression, anxiety and behavioural disorders being the leading causes of illness and disability among adolescents[2]. The

continuously rising prevalence of AD is also of a great concern as demonstrated by the Global Burden of Disease report, which reflects a continuous increase in the last two decades between 2009 and 2019[3]. People suffering from ADs typically endure intense levels of fear or distress disproportionate to real events. ADs affect both adults and children. The anxiety/fear occupying the daily lives of these people affects their behaviors, thoughts, and

---

[1]Department of Biomedical Engineering and Biotechnology, Khalifa University of Science and Technology, Abu Dhabi, UAE. [2]Center for Biotechnology (BTC), Khalifa University of Science and Technology, Abu Dhabi, UAE. [3]Biomedical and Mobile Health Technology Lab, ETH Zurich, Zurich, Switzerland. [4]Department of Electrical Engineering and Information Technology, Karlsruhe Institute of Technology (KIT), Karlsruhe, Germany. [5]ESAT - STADIUS Center for Dynamical Systems, Signal Processing and Data Analytics, Department of Development and Regeneration, KU Leuven, Leuven, Belgium. [6]Department of Medical Sciences, Khalifa University of Science and Technology, Abu Dhabi, UAE. [7]Institute of Health Care Engineering with European Testing Center of Medical Devices, Graz University of Technology, Graz, Austria. [8]These authors contributed equally: Mohamed Elgendi, Karmen Markov. ✉e-mail: mohamed.elgendi@ku.ac.ae; carlo.menon@hest.ethz.ch

physical health, often hindering their ability to function normally[4]. Several studies indicate a bidirectional relationship between anxiety disorders and cardiovascular diseases, i.e., anxiety disorders may be both causes and consequences of cardiovascular diseases[5-8]. The diagnosis of ADs is generally conducted in a clinic through a series of interviews and questionnaires. However, only one in four patients are reported to receive treatment for ADs due to lack of awareness, lack of competent or accessible mental health services, as well as lack of trained health care providers[1]. A conventional appointment with a practitioner at a clinic often results in delayed diagnosis, which can potentially lead to more severe illness[9]. Importantly, as mental health remains a stigma in many countries around the world, much less attention and resources are devoted to diagnostic and treatment tools for AD as comparted to other medical conditions. While ADs are clinically diagnosed through structured assessments, experimentally induced anxiety in healthy individuals is often used to examine associated physiological responses. These controlled paradigms help uncover autonomic and physiological changes linked to anxiety. Understanding these responses is essential for developing objective, technology-driven screening methods.

Today's rapidly emerging wearable device (WD) technology provides an opportunity for a paradigm shift in precision medicine, including mental health applications, such as AD. This includes using technology-driven tools as effective, and affordable means for screening for ADs beyond traditional clinical settings. Commonly, individuals with ADs experience a wide range of physical symptoms spanning multiple physiological systems, including cardiovascular symptoms (i.e., accelerated heart rate (HR), palpitations, arrhythmia), and chest pain, respiratory symptoms (i.e., choking sensation, the sensation of a lump in the throat), and dyspnea, and neuro-muscular responses (i.e., stiffness, paresthesia, contractures, muscle tension, weakness, and fatigue)[5,10]. Current wearable technologies can quantify these changes and provide continuous, non-invasive monitoring capabilities in real time by leveraging various physiological signals, such as electrocardiography (ECG), respiratory signals (RSP), electrodermal activity (EDA), or photoplethysmography (PPG). More specifically, biosignal analysis provides the potential of detecting ADs by analyzing the different signals and identifying features and patterns depicting anomalies. Multimodal WDs integrating various measurement technologies can capture diverse physiological signals which significantly enhances reliability and accuracy. Additionally, implementing multimodal or multiplexed analysis techniques leads to fewer false positives and simultaneously provides multiple output signals that correlate to a particular physiological or pathophysiological state for active calibration and correction[11]. Importantly, WDs provide people with the ability to detect anxiety in its early stages and seek timely medical treatment, thus reducing the corresponding disease morbidity, mortality, and subsequent healthcare costs.

ECG and PPG features, particularly heart rate variability (HRV), are widely used to detect anxiety disorders (ADs)[12]. ECG signals indicate the timing and rhythm of heartbeats by capturing the heart's electrical activity with high fidelity, enabling accurate derivation of HR and HRV metrics. However, ECG typically requires multi-electrode setups (e.g., chest or limb placement), which can limit wearability. In contrast, PPG captures peripheral blood volume changes using optical sensors, commonly worn at the wrist. Light is emitted into the skin, and the amount of light absorbed or reflected varies with blood flow during the cardiac cycle. PPG is widely used to derive cardiovascular parameters such as HR and oxygen saturation. While it offer better wearability, it is also more prone to motion artifacts and signal degradation under poor perfusion. Comparative studies show that PPG- and ECG-derived HRV metrics correlate well under ideal conditions, though PPG may be less reliable in ambulatory or real-world settings[13,14]. Despite decades of research on ECG-derived HRV, its utility in anxiety detection remains debated[15]. Respiration cycles involve inspiration (air inflow, diaphragm contraction, lung volume increase) and expiration (air outflow, diaphragm relaxation, lung volume decrease), producing measurable thoracoabdominal motion. Respiratory dysregulations, such as breath-to-breath respiratory instability, frequent sighing, are also common characteristics of ADs[16]. Breath rate generally increases under anxiety and

decreases under relaxation. Several studies indicated that ECG and RSP features were closely associated with anxiety and stress detection[15,17]. EDA reflects changes in skin conductance resulting from sweat gland activity, which is modulated by the autonomic nervous system. These changes occur in response to emotional arousal, stress, or cognitive load, making EDA a valuable marker for anxiety detection[18,19]. Some studies[20-22] also incorporate skin temperature (SKT) alongside EDA or PPG, as it may capture stress-related peripheral vasoconstriction. However, systematic assessments of these biosignals for anxiety detection are limited. Prior reviews[23-26] often focus more broadly on mental health rather than anxiety specifically, lack pooled performance comparisons across modalities or do not systematically assess study limitations.

This review examines 26 studies utilizing ECG, RSP, EDA, and PPG signals and their combinations, comparing single- and multimodal approaches. We analyze pooled accuracy, highlight the most frequently studied features and classifiers, and provide insights into the clinical potential of biosignal-based WDs for anxiety detection. We show that multimodal approaches achieve higher accuracy than single-signal methods, and that ECG is the most reliable individual signal. However, most studies are small and heterogeneous, limiting firm conclusions about the superiority of any one signal. Overall, wearable devices show strong potential for anxiety detection, but larger, standardized studies are needed to establish their role in clinical and everyday settings.

## Methods
### Search strategy and study eligibility
This systematic review was registered with the International Prospective Register of Systematic Reviews (PROSPERO, registration number CRD42025638476). Relevant peer-reviewed publications with full English text were selected in this review according to Preferred Reporting Items for Systematic Reviews and Meta-Analyses (PRISMA) guidelines[27]. Papers on anxiety detection using different signal inputs (ECG, RSP, EDA, PPG, and their combinations) were identified and reviewed. In the identification phase (see Fig. 1), three databases (Pubmed, IEEE, Scopus) were searched based on the following terms: (anxiety detection OR anxiety assessment) AND (electrocardiograph OR respiration signal OR respiratory signal OR breath OR EDA OR electrodermal activity OR PPG OR photoplethysmography) AND (wearable). While the search focused on ECG, RSP, EDA, and PPG, studies employing multimodal approaches that included additional signals (e.g. SKT or electromyography) alongside these core modalities were also included.

### Inclusion and exclusion criteria
This review selected studies published between 01 Jan 2014 and 01 Jan 2025, regardless of the study design and the country where the study was conducted. Before the screening, duplicate records were discarded from the identified papers based on the terms mentioned earlier. In the screening phase, the following inclusion criteria were applied: (1) Use of ECG, RSP, EDA, or PPG signals, or combinations, and their features used to detect anxiety; (2) Clear quantification/description of performance of methods and signals; (3) Use of WDs to obtain noninvasive biosignals; (4) Studies conducted in adults. Review articles and publications not written in English were excluded.

### Data extraction
The data extraction process focused on collecting comprehensive information from each publication to enable meaningful comparisons, including signal modalities utilized, anxiety induction methods, anxiety measurement tools, WDs used and their placement, sample sizes of the studies, extracted features, best-performing machine learning methods, validation methods for machine learning models, and evaluation metrics. While all reported metrics (e.g., precision, recall, F1-score) were extracted, accuracy (ACC) was used as the primary metric to ensure consistency, due to being the most consistently reported. However, it is important to note that ACC can be biased when class distributions are imbalanced, as it does not account for

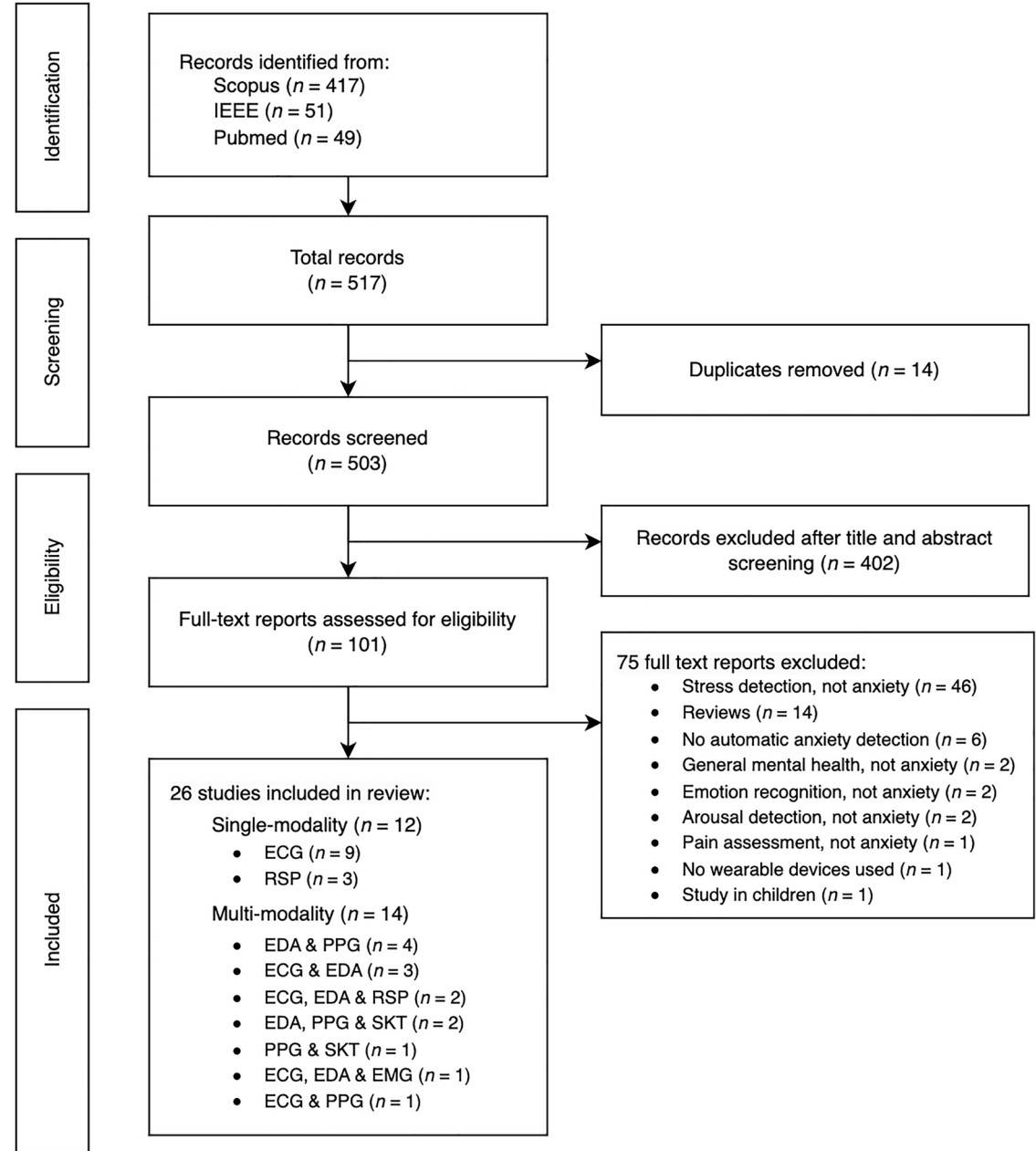

**Fig. 1 | PRISMA diagram showing study selection process.** The diagram illustrates the study identification, screening, eligibility assessment, and inclusion process. A total of 517 records were retrieved from Scopus, IEEE, and PubMed, of which 14 duplicates were removed. After title and abstract screening, 402 records were excluded. Of 101 full-text reports assessed, 75 were excluded for reasons including stress detection (not anxiety), reviews, absence of automatic anxiety detection, non-anxiety outcomes, non-wearable devices, or studies in children. Finally, 26 studies were included in the review. ECG electrocardiogram, EDA electrodermal activity, EMG electromyography, PPG photoplethysmography, RSP respiratory signal, SKT skin temperature.

false positives or false negatives in the minority class. Metrics like F1-score or Matthews correlation coefficient (MCC) better address this by incorporating both precision and recall, providing a more balanced view of model performance under class imbalance.

ACC is calculated as follows (see Eq. (1)):

$$ACC = \frac{TP + TN}{TN + TP + FN + FP},\qquad(1)$$

where:
- TP (True Positives) denotes the number of anxiety cases correctly identified and classified,
- TN (True Negatives) denotes the number of non-anxiety cases correctly classified,
- FN (False Negatives) represents the number of anxiety cases incorrectly classified as non-anxiety, and
- FP (False Positives) indicates the number of non-anxiety cases incorrectly classified as anxiety.

For studies that did not explicitly report accuracy, we estimated it using the provided TP, TN, FN, and FP values from that study. This estimated accuracy, calculated using the equation mentioned above, is denoted by *ACC*. This is distinct from ACC, which indicates that the accuracy was directly reported in the study.

## Pooled accuracy calculation

To determine the pooled accuracy across various datasets or studies, we employed the Cochrane formula[28,29] for weighted averages. In this approach, the accuracy of each study is weighted by its sample size, and the sum is then divided by the aggregate sample size from all the studies. The equation for pooled accuracy is as follows:

$$\text{Pooled ACC} = \frac{\sum_{i=1}^{n}(N_i \times \text{ACC}_i)}{\sum_{i=1}^{n} N_i}, \tag{2}$$

where:

- $N_i$ is the sample size of study $i$,
- $\text{ACC}_i$ is the accuracy of study $i$, and
- $n$ is the total number of studies.

## Statistics and reproducibility

Statistical analyses focused on ACC as the primary metric. Details of ACC and pooled ACC calculations, including weighting by sample size, are provided in the sections Data extraction and Pooled accuracy calculation. Reproducibility was ensured by applying predefined inclusion and exclusion criteria, and by extracting study characteristics systematically using a standardized template. Sample sizes were taken directly from each included study, and each study was considered one independent unit of analysis (replicate) in the synthesis. A completed PRISMA checklist is provided in the Supplementary Material to document compliance with reporting standards.

## Limitation assessment criteria

The limitations of each included study were systematically assessed across eight predefined categories: population representation (L1), sample size (L2), study completion rate (L3), selective reporting (L4), outcome measurement validity (L5), algorithmic transparency (L6), validation robustness (L7), and reproducibility (L8). Each category was assigned a level (low, moderate, serious, or could not be determined) based on predefined criteria. For example, population representation (L1) was evaluated based on demographic diversity, with homogeneous samples (e.g., single gender or narrow age groups) rated as serious and diverse populations rated as low. Sample size (L2) thresholds were set at <30 participants (Serious), 30–99 participants (moderate), and ≥100 participants (low). Study completion rate (L3) considered dropout documentation, while selective reporting (L4) evaluated comprehensiveness in presenting evaluation metrics and subgroup analyses. Outcome measurement validity (L5) focused on whether validated tools or unvalidated self-reports were used for anxiety assessment. Algorithmic transparency (L6) evaluated whether key methodological details, such as extracted features, classifier settings, and preprocessing steps, were sufficiently described. Validation robustness (L7) examined the use of robust techniques such as leave-one-subject-out (LOSO) cross-validation (CV) or external datasets, with weaker methods (e.g., training/testing overlap) rated as serious. Finally, reproducibility (L8) was assessed based on the availability of datasets, code, and tools for independent replication. This framework, developed for the present study, provided a consistent framework to evaluate limitations across studies.

## Results

### Study selection

As shown in Fig. 1, we identified 517 records from three databases: Scopus ($n = 417$), IEEE ($n = 51$), and PubMed ($n = 49$). Upon removing 14 duplicates, 503 studies remained for screening. Following title and abstract screening, 402 studies were excluded. The remaining 101 full-text reports were assessed for eligibility, leading to the exclusion of additional 75 studies based on the following reasons: stress detection, not anxiety ($n = 46$), review articles ($n = 14$), no automatic anxiety detection ($n = 6$), general mental health status, not anxiety ($n = 2$), emotion recognition, not anxiety ($n = 2$), arousal detection, not anxiety ($n = 2$), pain assessment, not anxiety ($n = 1$), no use of WDs ($n = 1$), and studies conducted in children ($n = 1$).

Ultimately, 26 studies were included in the review. Of these, 12 studies used single-modality signals, including ECG ($n = 9$)[30–38], RSP ($n = 3$)[39–41] The remaining 14 studies employed multi-modality approaches or reported results for multiple signals: EDA & PPG ($n = 4$)[18,19,42,43], ECG & EDA ($n = 3$)[44–46], ECG, EDA & RSP ($n = 2$)[47,48], EDA, PPG & SKT ($n = 2$)[20,21], PPG & SKT ($n = 1$)[22], ECG, EDA & EMG ($n = 1$)[49], and ECG & PPG ($n = 1$)[50]. The study design and data analysis workflow for the 26 included studies are summarized in Fig. 2. By providing an overview of the key methodological steps, this figure highlights the diversity in study designs and approach among the 26 articles, which is further elaborated in the subsequent sections.

### Included studies

The 26 studies reviewed (summarized in Supplementary Data 1 and Supplementary Data 2) employed a variety of biosensors, datasets, anxiety induction and measurement methods, data preprocessing and machine learning approaches, resulting in a wide range of reported accuracies. The majority of these studies were conducted in controlled lab environments, with only four studies[37–39,41] utilizing real-world settings. For single biosensors, classification accuracies ranged from 42.53% for EDA, reported by Lee et al.[18], to 99.95% for ECG by Baygin et al.[30]. For multisensor recordings, 99.84% was the highest reported accuracy by Sinche et al.[50], based on ECG and PPG sensor recordings, while Lee et al.[18] reported the lowest accuracy of 49.26% when combining PPG, EDA, and EEG recordings. The differences in accuracies have to be viewed with caution as they are derived from different recording devices (e.g., Empatica E4[18], SS2LB ECG module[30], AD8232 and MAX30100 sensors[50]), sample sizes, machine learning algorithms, and study conditions, which should be considered when comparing results. Incorporating contextual features, such as demographic and situational data, has been shown to enhance classification accuracies in anxiety detection. For instance, Jain & Kumar[44] utilized the Wearable Stress and Affect Detection (WESAD) dataset[51] to classify anxiety levels in 15 participants using ECG and EDA signals from a chest-worn RespiBAN device. They achieved accuracies of 89.80% for ECG, 85.90% for EDA, and 96.70% for combined signals, which further increased to 97.30% with the addition of contextual features. Similarly, Nath & Thapliyal[19] analyzed data from 41 older adults using wristband sensors to capture EDA and PPG signals during the Trier Social Stress Test (TSST)[52], reporting accuracies of 89.00% for EDA and 78.00% for PPG. The inclusion of contextual features improved these accuracies to 92.00% for EDA and 83.00% for PPG. However, several studies reported accuracies exceeding 90% also without contextual features. For example, Baygin et al.[30], Sinche et al.[50], Shaukat-Jali et al.[21], Zhou et al.[45], Banerjee et al.[40], Jain & Kumar[44], Padmaja et al.[32], Di Tecco et al.[42], Petrescu et al.[43], Haritha et al.[41], Tripathy et al.[31], Vaz et al.[49], Wen et al.[35] achieved notable results. Among these, Vaz et al.[49] achieved an accuracy of 84.50% using ECG, EDA, and EMG signals from the WESAD dataset[51]. To address the issue of class imbalance, Synthetic Minority Oversampling Technique (SMOTE) was applied. This led to a notable increase in accuracy to 92.00%. SMOTE's ability to enhance performance highlights the importance of addressing dataset imbalances in developing robust anxiety detection models.

### Datasets employed across studies

The reviewed studies employed a variety of datasets for anxiety detection. A significant proportion, 16 studies[18–22,32,35–43,50], collected data specifically for their research. The WESAD dataset[51], used in three studies[44,45,49], comprises of data from 15 participants collected using ECG, EDA, and respiratory signals captured using a chest-worn RespiBAN device. Anxiety was induced through the TSST[52], and anxiety levels were categorized as low, moderate, or high based on self-reports from the 6-item State and Trait Anxiety Inventory (6-STAI)[53]. The dataset by Ihmig et al. 2020[54], used in four of the reviewed studies[33,46–48], includes ECG, respiratory (RSP), and skin conductance signals from 57 participants aged 18 to 40 years. The signals were recorded using a BITalino device during a protocol involving 16 spider-themed video clips presented in a virtual reality (VR) setting. Each session consisted of four 1 min spider-themed clips, followed by a 5-minute resting period.

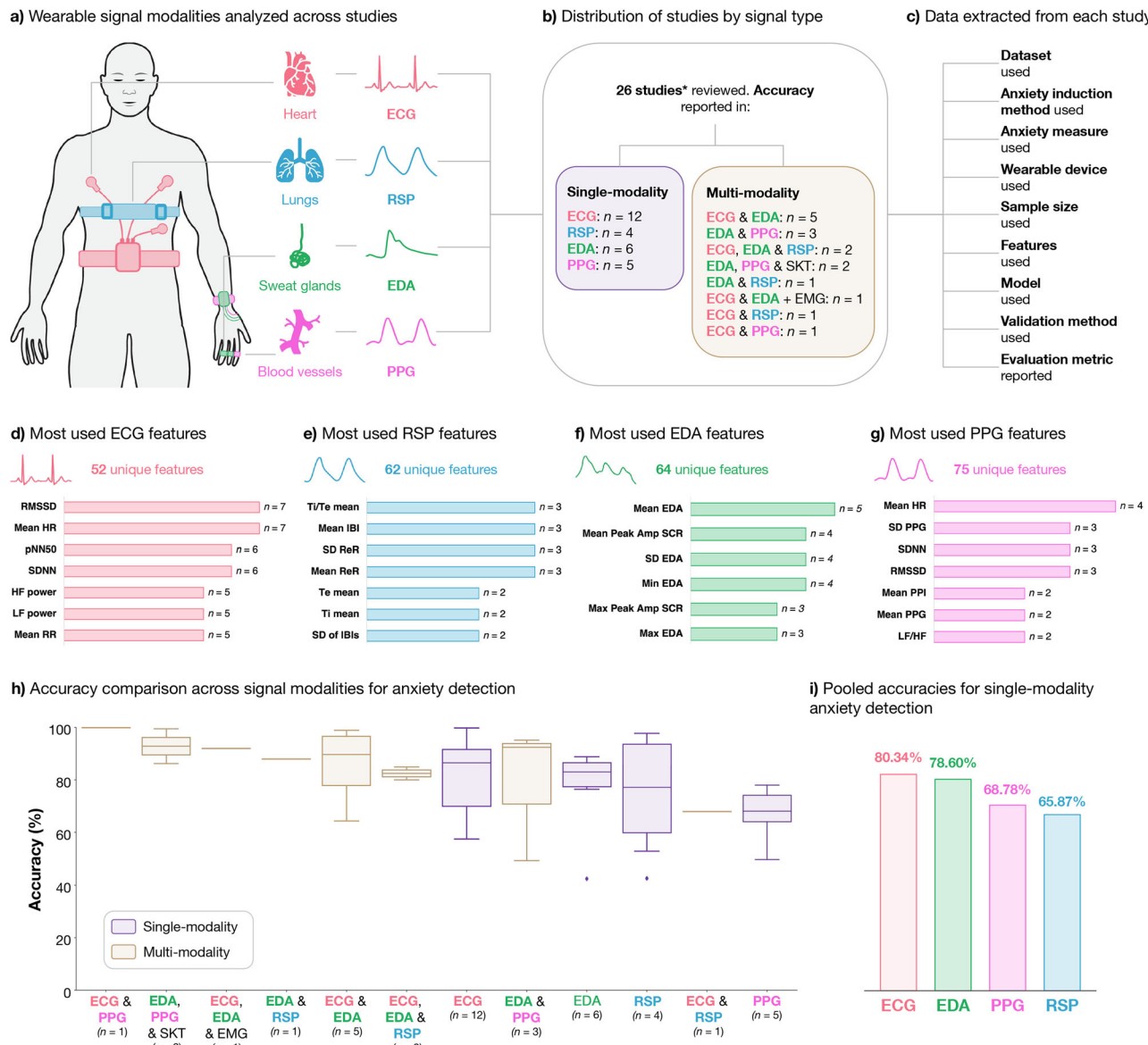

**Fig. 2 | Study design and data analysis workflow for wearable-based anxiety detection review.** The study design and data analysis workflow consisted of four steps: search and selection of studies, data extraction, feature analysis, and performance analysis. **Step 1: Search and selection of studies.** **a** shows the wearable signal modalities analyzed: ECG, RSP, EDA, and PPG. **b** presents the number of reviewed studies using each signal modality, including single- and multi-modality approaches. Some studies contributed to multiple categories; for example, Jain & Kumar[44] reported results using ECG alone, EDA alone, and ECG + EDA combined, and were counted in all relevant categories. **Step 2: Data extraction.** **c** highlights the elements extracted from each study, including dataset, features, model, validation, and evaluation metrics. **Step 3: Feature analysis.** **d**–**g** summarize the most commonly used features for ECG, RSP, EDA, and PPG, respectively, and their frequencies of use. **Step 4: Performance analysis.** **h** displays boxplots illustrating the distribution of accuracy results across different signal modalities, comparing single-modality and multi-modality approaches, while **i** presents pooled accuracies for single-modality anxiety detection, highlighting the overall performance of ECG, EDA, PPG, and RSP signals. **Note:** In (**b**), n refers to the number of independent studies using each signal modality (single- or multi-modality). In **d**–**g**, n refers to the number of independent studies that reported the use of each specific feature. In (**h**), n indicates the number of independent study results contributing to each boxplot. ECG electrocardiogram, EDA electrodermal activity, PPG photoplethysmography, RSP respiratory signal, HR heart rate, SDNN standard deviation of normal-to-normal intervals, RMSSD root mean square of successive differences, HF power high-frequency power, LF power low-frequency power, Mean RR mean R-R interval, Ti/Te inspiratory to expiratory time ratio, IBI inter-breath interval, ReR respiratory rate, SCL skin conductance level, SCR skin conductance response, PPI pulse interval, LF/HF low-frequency to high-frequency power ratio.

Participants rated their subjective arousal on a four-point scale ("1 = not at all" to "4 = strongly") after each clip. The dataset by Elgendi et al. 2022[55], used in two studies[30,31], includes ECG signals collected from 19 participants exposed to anxiety-inducing and non-anxiety video clips. Anxiety levels were measured using the Hamilton Anxiety Rating Scale (HAM)[56], and the signals were recorded using the SS2LB ECG module. The Anxiety Phases Dataset (APD)[57], used in Zhou et al. 2023[45], includes data from 52 participants who performed public speaking and bug-box tasks. Anxiety levels

were assessed using the Subjective Units of Distress Scale (SUDS)[58]. Signals were recorded using the Zephyr BioHarness 3.0 for chest ECG and the Grove-GSR Sensor for wrist EDA. Lastly, a undefined Kaggle dataset focusing on ECG signals was used in Tang et al. 2021[34].

**Anxiety induction methods employed across studies**

The studies reviewed here employed various anxiety induction methods, as illustrated in Fig. 3a. Public speaking tasks, which were the most common

**a) Anxiety induction methods**

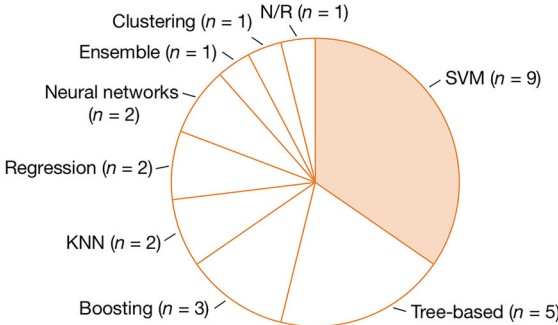

**b) Anxiety measurement methods**

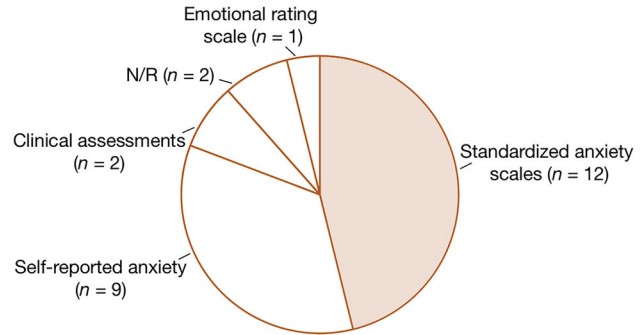

**c) Models used for anxiety detection**

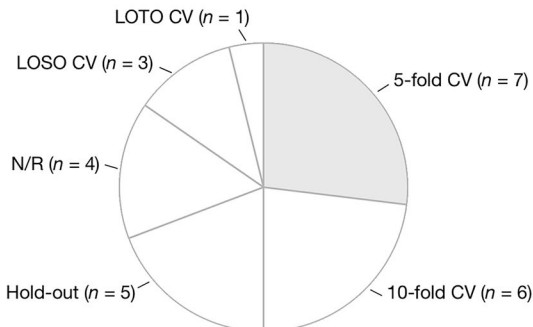

**d) Validation methods**

**Fig. 3 | Overview of anxiety detection methodologies across reviewed studies.**
**a** Anxiety induction methods: Proportions of methods used to induce anxiety in the studies. **b** Anxiety measurement methods: Proportions of methods used to measure anxiety across studies. **c** Models used for anxiety detection: Proportions of machine learning models employed for anxiety detection. **d** Validation methods: Proportions of strategies employed to validate models. **Note:** In all panels, *n* refers to the number of independent studies reporting the use of the respective method. LOTO CV leave-one-trial-out cross-validation, LOSO CV leave-one-subject-out cross-validation, N/R not reported, VR virtual reality, SVM support vector machines, KNN K-Nearest Neighbors, CV cross-validation.

method, were used in eight studies. These included the TSST[52], which involves public speaking and mental arithmetic task, used in five studies[19,35,44,45,49]; a public speaking task with a bug-box component[45], an impromptu speech task[21]; and a Virtual Reality Exposure Therapy (VRET) public speaking scenario[20]. VR-based anxiety induction methods were employed in five studies, including spider exposure[33,46–48] and exposure to heights[43]. Video-based methods were used in four studies, including general anxiety-inducing videos[30,31], a purposely edited horror movie trailer[42], and driving anxiety-inducing videos[18]. Other anxiety induction techniques included detecting anxiety in hospital shift work settings[37–39], pre- and post-test evaluations comparing anxious versus non-anxious states[50], baseline versus relaxation tasks with biofeedback for prenatal anxiety detection in pregnant women[22], an affective Pacman gaming task[59] to induce frustration[40], and an experimental task designed specifically for Social Anxiety Disorder (SAD)[36]. Additionally, two studies did not specify the anxiety induction methods used[32,34].

**Anxiety measurement tools employed across studies**
The reviewed studies utilized a range of tools to measure the levels of anxiety, encompassing both standardized scales and self-reported measures, as illustrated in Fig. 3b. Standardized anxiety scales were the most commonly used method, applied in 12 studies, including STAI[53] used in five studies[19,22,44,45,49], SUDS[58], wich was used in three[20,43,45], Cognitive Test Anxiety Scale[60] used in one[50], Social Phobia Screening Questionnaire (SPSQ)[61] used in one[21], HAM[56] used in two[30,31], and LSAS[62] used in two studies[21,36]. Self-reported anxiety levels were employed in nine studies, with methods ranging from self-reported anxiety onset[18,48], predominant emotion[42], and daily ratings on a 5-point scale[37–39] to general self-reported

anxiety levels[33,35,46,47]. Additionally, Wen et al.[35] employed audience-observed anxiety scores based on symptoms, such as sweating, trembling, blushing and avoiding eye contact as ground truths for labeling anxiety states. Clinical assessments were used in two studies to measure anxiety, including the Diagnostic and Statistical Manual of Mental Disorders (DSM-5)[63] diagnostic criteria in[36], and clinical psychiatric diagnoses in[41]. Emotional rating scales, such as Self-Assessment Manikin (SAM), were employed in a single study[40]. Two studies[32,34] did not report specific anxiety measurement tools, which limits the comparability of their findings. Overall, the studies highlight the diverse methods used to measure anxiety.

In terms of classification schemes, anxiety was labeled as binary (e.g., anxious vs. non-anxious) in 18 studies[18,19,21,32–35,37–41,45–50], while four studies used 3-class schemes (e.g., low/moderate/high)[22,42–44], and three studies used 4-class schemes (e.g., normal/mild/moderate/severe)[20,30,31]. One study did not report its labeling method[36]. Most studies that used standardized scales also used them to define class labels. However, in some cases, labels were derived from experimental context. For example, Vulpe-Grigorasi et al.[33], Gazi et al.[47], and Khullar et al.[48] assigned labels based on exposure to spider-related VR clips versus rest, despite collecting self-reported anxiety levels. Sinche et al.[50] used the Cognitive Test Anxiety Scale for assessment, but labeled data based on timing, pre-test as anxious and post-test as non-anxious. Banerjee et al.[40] used SAM ratings, but labeled gameplay windows as anxious or non-anxious depending on whether participants played a frustration-inducing or standard Pacman game. Finally, Shaukat-Jali et al.[21] employed LSAS-SR and SPSQ for anxiety assessment, but classification into anxious versus baseline was based on the timing of data collection relative to the impromptu speech task.

**Fig. 4 | Placement of ECG and RSP WDs in this study. a** Wearable ECG device. RA is right arm lead, LA is left arm lead and LL is left leg lead. **b** T-shirt ECG device. **c** Inductive RSP device. **d** Piezoelectric RSP device. **e** Wrist-mounted EDA and PPG device. **f** Finger-mounted EDA and PPG device. ECG electrocardiogram, EDA electrodermal activity, PPG photoplethysmography, RSP respiratory signal.

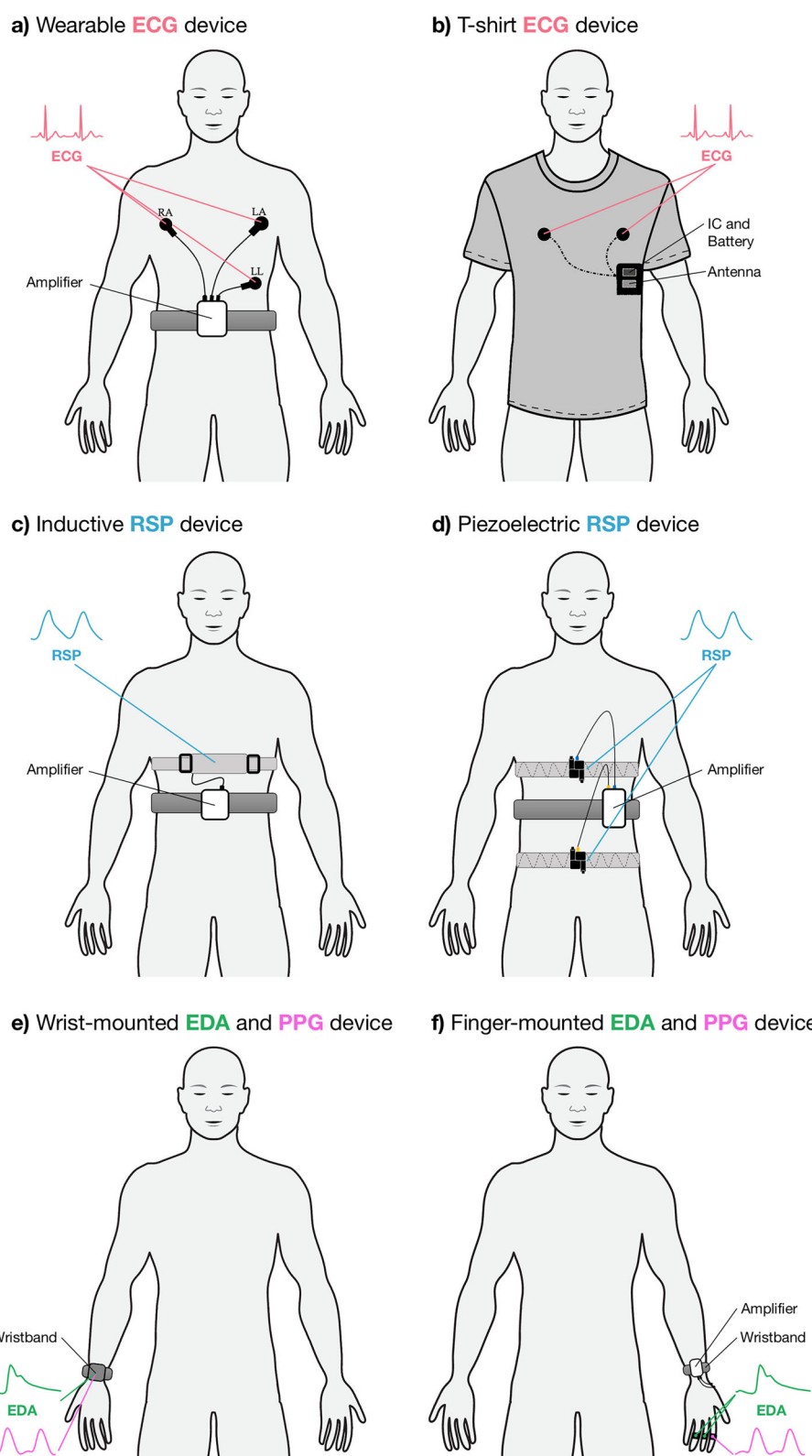

## Wearable devices employed across studies

The reviewed studies included here used various WDs with diverse configurations and placements to capture ECG, EDA, PPG, and RSP signals for anxiety detection. The sketches of the devices used are provided in Fig. 4, and the devices that each study used are listed in Supplementary Data 1 and Supplementary Data 2. Below, we group the

devices based on the signal modalities they recorded in the included studies.

Nine devices were employed to record single signal modalities in the studies reviewed. Out of these, five devices were used to record ECG signals: (1) the SS2LB ECG module[30,31]; (2) AD8232 sensor[50]; (3) the Biopac MP-45 system[32]; (4) the MP150 recorder[35]; (5) the Zephyr BioHarness 3.0[45]. The

SS2LB ECG module, used in the Elgendi et al.[55] dataset utilized by Baygin et al.[30] and Tripathy et al.[31] employed a Lead I configuration (right arm (RA) to left arm (LA), left leg (LL) as ground), designed for high-resolution recordings compatible with Biopac systems. The AD8232 sensor, used by Sinche et al.[50], consists of a compact analog front-end module placed on the left side of the chest with three electrodes configured in a standard Lead II setup. The Biopac MP-45 system, used by Padmaja et al.[32], featured an embedded CPU for data acquisition and supported various electrode types, though specific placements were not detailed. The MP150 multi-channel recorder, used by Wen et al.[35], collected ECG signals with electrodes placed on both wrists and the right ankle, sampling at 400 Hz, alongside skin conductance data. Lastly, the Zephyr BioHarness 3.0, used in the APD[57] dataset utilized by Zhou et al.[45], recorded chest-based ECG data with two disposable adhesive electrodes aligned to the breastbone, sampling at 250 Hz. Most of these devices, as illustrated in Fig. 4a, are attached to the subject with a strap. Two devices were employed to record RSP data: (1) Embletta[41] and (2) an undefined respiration belt[40]. The Embletta device (Fig. 4c), used by Haritha et al.[41], contains two elasticity belts with sinusoid conducting coils for thoracic-abdominal movement detection. One belt is placed on the chest, and the other is placed on the subject's abdomen, such that variations of the thoracic and abdominal volumes cause changes in the surface of the section and stretching of elasticity belts. This leads to variation of the self-inductance of the coils and the frequency of their oscillation, hence inducing a current[64]. Banerjee et al.[40] obtained the RSP using a respiration belt that measures the expansion of the abdomen[65] related to the quantity of inspired and expired air. One sensor was used to measure EDA signal, the Grove-GSR Sensor V1.2, utilized by Zhou et al.[45]. It measured EDA signals at a sampling rate of 50 Hz using electrodes placed on the index and ring fingers, with the device enclosed in a 3D-printed module secured on both wrists via Velcro straps. One sensor, the MAX30100 sensor, employed by Sinche et al.[50], recorded PPG signals from the students' index finger.

Five devices were employed to record multiple signal modalities in the studies reviewed: (1) the RespiBAN device[44,45,49] for recording ECG and EDA signals; (2) the OMSignal smartshirt[37–39] for ECG and RSP signals; (3) BiTalino[33,46–48] for ECG, EDA and RSP signals; (4) Empatica E4[18,20–22,44,45,49] for EDA and PPG signals; (5) Shimmer3 GSR+[42,43] also for EDA and PPG signals. The RespiBAN device, a chest-worn sensor capable of capturing ECG, EDA, EMG, RSP, body temperature, and acceleration, was used in the WESAD dataset[51] and employed by Jain and Kumar[44], Zhou et al.[45], and Vaz et al.[49]. The OMSignal smartshirt (Fig. 4b) used by Tiwari et al.[37–39] contains a close-fitting T-shirt and an OMBox. The T-shirt has sensors that are directly knit into the textile. It leverages conductive yarns with electromechanical properties. The OMBox is fastened to the T-shirt using snap buttons. A battery that lasts for 30 h of real-time usage powers the OMBox. The built-in Bluetooth module connects to the mobile phone and transfers data. As shown in Fig. 4d, the BITalino respiration belt, which was used in the dataset by Ihmig et al. 2020[54] and used in four studies[33,46–48] is strapped around the subject's chest for recording changes in thoracic movement during breathing. The Empatica E4 wristband (Fig. 4e), used in the WESAD dataset[51] utilized in three studies[44,45,49] and four further studies[18,20–22], is worn on the non-dominant wrist and records EDA at 4 Hz, PPG at 64 Hz, body temperature at 4 Hz, and three-axis acceleration at 32 Hz. The Shimmer3 GSR+ device (Fig. 4f), used in two reviewed studies[42,43], measures EDA using electrodes placed on the index and middle fingers, capturing PPG signals via an optical pulse probe on the ring finger and transmitting data in real-time via Bluetooth.

## Features employed across studies

For ECG signals, a total of 52 unique features were extracted across the reviewed studies (Fig. 5a). The mean HR[36,43–47,50] and RMSSD[33,37,38,44,45,47,50] were the most frequently utilized features, both used in seven studies. The other frequently used features included standard deviation of RR intervals (SDNN), also used in six studies[35,37,44,47,50]; as well as the percentage of successive RR intervals differing by more than 50 ms (pNN50), used in six

studies[35,37,38,47,49,50]. Additionally, low- (LF) and high-frequency (HF) power of RR intervals were extracted in five studies[34,35,37,38,45], and the mean RR interval was also employed in five studies[35,37,38,44,47] (Fig. 5b).

For PPG signals, 75 unique features were extracted across the reviewed studies (Fig. 5a). The most frequently used feature was the mean HR, which appeared in four different studies[19,20,22,42]. The standard deviation of the PPG signal (SD PPG)[20,22,42] and the standard deviation of the NN intervals (SDNN)[20,22,50], as well as the root mean square of successive differences of NN intervals (RMSSD)[20,22,50] were reported in three studies. Other extracted features that were reported in more than one study included the mean inter-beat interval (Mean PPI)[20,50], the mean of the PPG signal (Mean PPG)[20,22], and the low-frequency to high-frequency power ratio (LF/HF)[18,22](Fig. 5c).

For EDA signals, 62 unique features were extracted across the reviewed studies (Fig. 5a). The most frequently employed feature was the mean of the EDA signal (Mean EDA), utilized in five studies[18,20,44–46]. Other commonly reported features included the mean peak amplitude of the phasic skin conductance response (SCR), the phasic component of EDA (Mean Peak Amp SCR), used in four studies[19,20,44,47], and the standard deviation of the EDA signal (SD EDA), also reported in four studies[18,20,44,45]. The minimum (Min EDA) and maximum (Max EDA) values of the EDA signal were each utilized in three studies[18,20,44]. Additionally, the maximum amplitude of the detected peaks in the SCR (Max Peak Amp SCR) was also extracted in three studies[19,20,49] (Fig. 5d).

For RSP signals, 62 unique features were extracted across the reviewed studies (Fig. 5a). The most commonly utilized features included the mean ratio of inspiratory time to expiratory time (Ti/Te mean)[39,40,47], and also reported in the same three studies[39,41,47], the mean breath-by-breath inter-breath intervals (Mean IBI), the mean respiratory rate (Mean ReR), and the standard deviation of respiratory rate (SD ReR). Other notable features found included the mean inspiratory time (Ti mean) and mean expiratory time (Te mean), both reported in two studies[40,47], as well as the standard deviation of breath-by-breath interbreath intervals (SD of IBIs), reported in two studies[39,47](Fig. 5e).

## Models and validation methods

The studies reviewed here employed a variety of machine learning models to detect anxiety as illustrated in Fig. 3c. Among these, Support Vector Machines (SVMs) were the most commonly used models, employed in nine studies[20,22,30,35,37–39,41,50]. Tree-based models were also widely used, with five studies using methods such as Random Forest (RF)[19,47], Decision Tree (DT)[36], ET[48], and Bagged Trees[46]. Boosting models were used in three studies, including Adaptive Boosting[49], Extreme Gradient Boosting (XGB)[31], and Gradient Tree Boosting (GTB)[44]. K-Nearest Neighbors (KNN) models were applied in two studies[21,42] and regression models were also used in two[18,43]. Two other studies explored neural network architectures, including using a Multi-Layer Perceptron (MLP) model[40] and a 1D Convolutional Neural Network (CNN)[33].Other methods included ensemble techniques, combining SVM, Light Gradient Boosting Machine (LGBM), RF, and XGB[45], and clustering-based methods, such as a Density-Based Spatial Clustering of Applications with Noise (DBSCAN)-enhanced algorithm[34]. Finally, one study[32] did not report the model used.

The most common validation approach was 5-fold CV, used in seven studies[35,37–39,45,48,49], followed by 10-fold CV in six studies[20,21,30,40,46,50]. Hold-out validation was used in five studies[19,31,33,42,43]. LOSO CV was applied in three studies[22,44,47], while Leave-One-Task-Out (LOTO) CV was employed in one study[18]. Four studies[32,34,36,41] did not specify the validation methods used (Fig. 3d).

## Pooled and mean accuracies

The studies reviewed in this article reported varying levels of accuracy across single-modality and multi-modality approaches for capturing and analyzing various physiological signals reflecting AD, as summarized in Fig. 6a, b, and Table 1. It is important to note that mean and pooled accuracies were calculated only from studies which reported results on accuracy, or where accuracy was possible to calculate. For example, the results from Bao et al.[22]

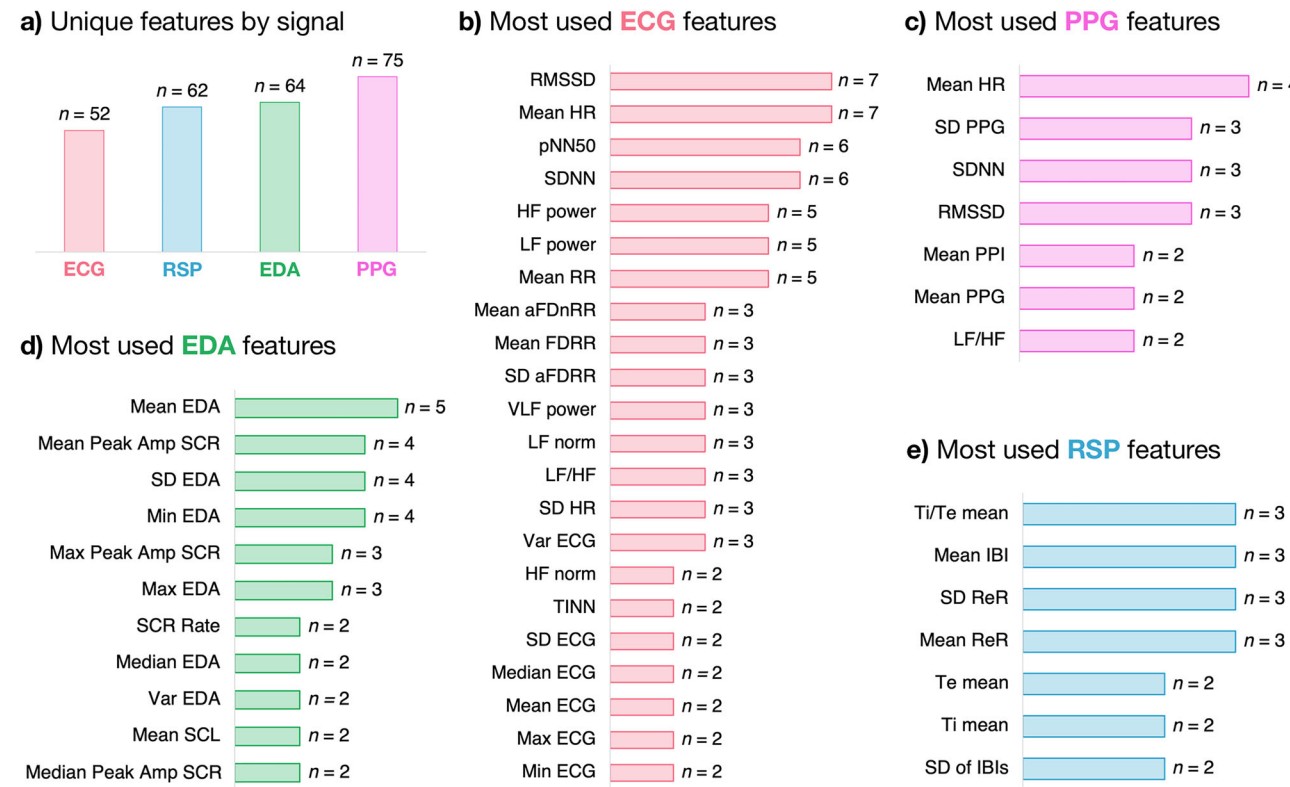

**Fig. 5 | Overview of features extracted from physiological signals across reviewed studies for anxiety detection. a** Unique features by signal: Number of unique features extracted from each signal modality (ECG, RSP, EDA, and PPG). **b** Most used ECG features: ECG features reported more than once across studies, highlighting HRV features (e.g., Mean HR, pNN50, and SDNN). **c** Most used RSP features: RSP features reported more than once across studies, focusing on respiratory timing and interval measures (e.g., Ti/Te mean, Mean IBI). **d** Most used EDA features: EDA features reported more than once across studies, highlighting amplitude and peak-related metrics (e.g., Mean EDA, Mean Peak Amp SCR). **e** Most used PPG features: PPG features reported more than once across studies, emphasizing heart rate and variability metrics (e.g., Mean HR, SD PPG). Note: In (**a**), $n$ indicates the number of unique features identified across studies for each signal modality. In (**b**–**e**), $n$ indicates the number of independent studies reporting the use of each specific feature. ECG features: Mean HR mean heart rate, pNN50 percentage of successive RR intervals differing by more than 50 ms, SDNN standard deviation of RR intervals, RMSSD root mean square of successive RR interval differences, HF power high-frequency power of heart rate variability, LF power low-frequency power of heart rate variability, Mean RR mean R-R interval, Mean aFDnRR mean of the absolute values of the first differences of normalized RR intervals, Mean FDRR mean of the first differences of RR intervals, SD aFDRR standard deviation of the absolute values of the first differences of RR intervals, VLF power very low-frequency power of RR intervals, LF norm normalized low-frequency power, LF/HF ratio of low-frequency to high-frequency power, SD HR standard deviation of heart rate, Var ECG variance of the ECG signal, HF norm normalized high-frequency power, TINN triangular interpolation of NN interval histogram, SD ECG standard deviation of the ECG signal, Median ECG median of the ECG signal, Mean ECG mean of the ECG signal, Max ECG maximum amplitude of the ECG signal, Min ECG minimum amplitude of the ECG signal. PPG features: mean HR mean heart rate, SD PPG standard deviation of the PPG signal, Mean PPI mean pulse interval, Mean PPG mean of the PPG signal, LF/HF ratio of low-frequency to high-frequency power. EDA features: mean EDA mean electrodermal activity amplitude, Mean Peak Amp SCR mean amplitude of skin conductance response peaks, SD EDA standard deviation of the EDA signal amplitude, Min EDA minimum amplitude of the EDA signal, Max Peak Amp SCR maximum amplitude of SCR peaks, Max EDA maximum amplitude of the EDA signal, SCR Rate rate of skin conductance response peaks per unit time, Median EDA median of the EDA signal amplitude, Var EDA variance of the EDA signal amplitude, Mean SCL mean of the tonic component (skin conductance level) of the EDA signal, Median Peak Amp SCR median of the amplitudes of detected peaks in the SCR. RSP features: Ti/Te mean mean inspiratory to expiratory time ratio, Mean IBI mean inter-breath interval, SD ReR standard deviation of the respiratory rate, Mean ReR mean respiratory rate, Te mean mean expiratory time, Ti mean mean inspiratory time, SD of IBIs standard deviation of inter-breath intervals.

which provided F1 scores, but did not report accuracy, were excluded from our calculations. Among single-modality results, ECG exhibited the highest pooled accuracy at 80.43%, with a mean accuracy of 81.10% (±14.70%), based on results from 12 different studies[30–38,44,47,50], with accuracies ranging from 57.65[37] to 99.95%[30]. EDA followed with a pooled accuracy of 79.34% and a mean accuracy of 76.92% (±17.46%), based on results from six studies[18–21,44,47], with accuracies ranging from 42.53[18] to 89.00%[19]. PPG signals demonstrated a pooled accuracy of 68.78%, with a mean accuracy of 66.83% (±10.94%), based on five studies[18–22,50], ranging from 49.75[18] to 78.00%[19]. RSP, though showing the lowest pooled accuracy at 65.87%, had a wider variation in results, with a mean accuracy of 76.40% (±22.05%) from four studies[39–41,47], ranging from 53.00[47] to 97.90%[40]. Overall, single-modality approaches revealed a pooled accuracy of 76.98% and a mean accuracy of 76.83% (±15.88%) across 27 reported results from 17 different studies.

In studies with multi-modality approaches, ECG & PPG achieved the highest accuracy of 99.84%, although it was based on a single study[50]. ECG, EDA & EMG achieved an accuracy of 92.00%, also based on a single article[49]. The combination of EDA, PPG & SKT demonstrated a pooled accuracy of 90.07% and a mean accuracy of 92.89% (±9.32%), based on two studies, 86.30[20] and 99.48%[21], respectively. The combination of EDA & RSP yielded an accuracy of 88.00%, also derived from a single study[47]. ECG, EDA & RSP achieved a pooled accuracy of 82.46% and a mean accuracy of 82.50% (±3.54%), based on two studies, achieving 80.0[48] and 85.00%[47], respectively. ECG & EDA, a commonly used combination, reported a pooled accuracy of 80.92% and a mean accuracy of 85.60% (±14.35%) across five results from four studies[44–47]. Accuracies ranged from 64.50 to 99.00%, both provided by Zhou et al.[45], who trained their model on two distinct datasets, APD[57] (64.50%) and WESAD[51] (99.00%). This wide range illustrates how

## a) Accuracy results across studies by signal(s) used

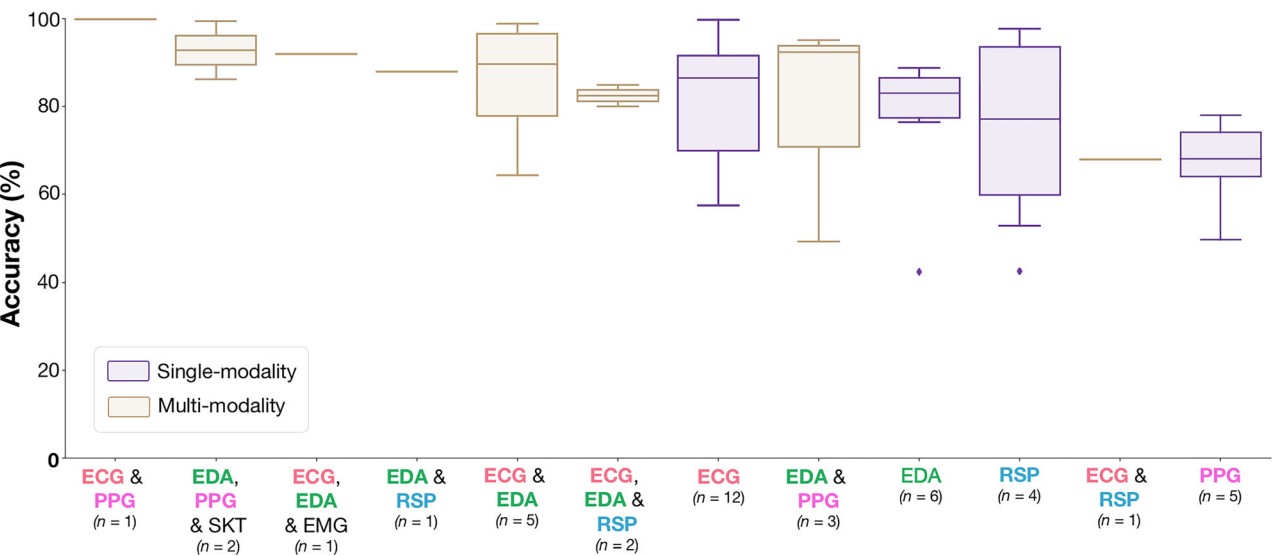

## b) Pooled accuracies by signal(s) used

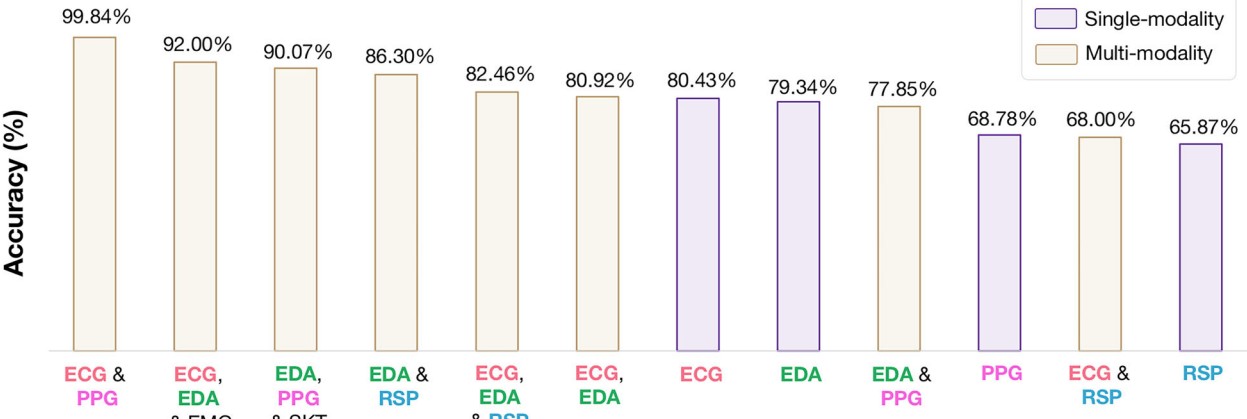

**Fig. 6 | Accuracy results for anxiety detection across studies by signal(s) used.**
**a** Accuracy results by signal(s) used: Boxplots illustrate the distribution of reported accuracies across studies, where the centre line indicates the median, the box represents the interquartile range (IQR), and whiskers show the minimum and maximum values. Categories with fewer than three studies are shown as individual points without boxplots. Single-modality results are highlighted in green, while multi-modality results are shown in purple. **b** Pooled accuracies for anxiety

detection: Bar chart displaying pooled accuracy values for each signal modality or combination, calculated as weighted averages based on the number of studies reporting each result. **Note:** In (**a**), *n* indicates the number of independent study results contributing to each boxplot. ECG electrocardiogram, EDA electrodermal activity, PPG photoplethysmography, RSP respiration, SKT skin temperature, EMG electromyography.

dataset-specific factors, such as the number of participants (52 for APD versus 15 for WESAD) and experimental protocols (public speaking and bug-box tasks for APD versus TSST for WESAD), can significantly influence model performance. EDA & PPG achieved a pooled accuracy of 77.85% and a mean accuracy of 78.95% (±25.75%) based on three studies[18,42,43], ranging from 49.26%[18] to 95.22%[42]. Lastly, the combination of ECG & RSP yielded an accuracy of 68.00% based on a single study[47]. Multi-modality approaches collectively achieved a pooled accuracy of 81.98% and a mean accuracy of 85.22% (±14.29%) across 16 reported results from 12 different studies.

### Assessment of study limitations
A systematic assessment of 26 studies revealed varying levels of limitations across eight predefined categories (Fig. 7). The most prevalent serious limitations were found in population representation (L1) (38%),

sample size (L2) (38%), and reproducibility (L3) (58%), reflecting challenges in ensuring diverse, generalizable findings as well as availability of datasets and code for replication. In contrast, study completion rate (L3) and algorithmic transparency (L6) were frequently rated as Low (81% and 85%, respectively), suggesting consistent reporting of dropout rates and well-explained methods. Selective reporting (L4) and validation robustness (L7) were primarily rated moderate (77% and 65%, respectively), highlighting common gaps in reporting secondary evaluation metrics, such as sensitivity, specificity, and F1 scores. Furthermore, while many studies relied on standard validation techniques, such as 5-fold or 10-fold cross-validation or hold-out validation, more rigorous methods, including LOSO CV or validation on independent datasets, were frequently absent. Outcome measurement validity (L5) often relied on partially validated methods, such as self-reports (54%), raising concerns about reliability.

**Table 1 | Summary of pooled and mean accuracies across signal modalities for anxiety detection**

| Signal | Pooled accuracy (%) | Mean accuracy* (%) | Number of results |
|---|---|---|---|
| *Single-modality* | | | |
| ECG | 80.43 | 81.10 ± 14.70 (57.65–99.95) | 12 |
| EDA | 79.34 | 76.92 ± 17.46 (42.53–89.00) | 6 |
| PPG | 68.78 | 66.83 ± 10.94 (49.75–78.00) | 5 |
| RSP | 65.87 | 76.40 ± 22.05 (53.00–97.90) | 4 |
| Single-modality | 76.98 | 76.83 ± 15.88 (42.53–99.95) | 27 |
| *Multi-modality* | | | |
| ECG & PPG | 99.84 | 99.84 | 1 |
| ECG, EDA & EMG | 92.00 | 92.00 | 1 |
| EDA, PPG & SKT | 90.07 | 92.89 ± 9.32 (86.30–9.48) | 2 |
| EDA & RSP | 88.00 | 88.00 | 1 |
| ECG, EDA & RSP | 82.46 | 82.50 ± 3.54 (80.00–85.00) | 2 |
| ECG & EDA | 80.92 | 85.60 ± 14.35 (64.50–99.00) | 5 |
| EDA & PPG | 77.85 | 78.95 ± 25.75 (49.26–95.22) | 3 |
| ECG & RSP | 68.00 | 68.00 | 1 |
| Multi-modality | 81.98 | 85.22 ± 14.29 (49.26–99.84) | 16 |

*Mean accuracy is reported as mean ± standard deviation (minimum–maximum). *ECG* electrocardiogram, *EDA* electrodermal activity, *PPG* photoplethysmography, *RSP* respiratory signal.

## Discussion

We conducted a comprehensive review of the emerging evidence regarding the use of physiological signals, including ECG, respiratory signals, EDA, and PPG, to detect anxiety. Most studies selected for this review primarily focused on affective or emotional states rather than clinical populations with medically diagnosed anxiety. Though this growing research area is still in its early stages, the findings from this review may offer preliminary insights into its clinical impact and lay the groundwork for subsequent studies. Below, we discuss the implications of our findings and potential applications.

Comprising of the sympathetic nervous system (SNS) and the parasympathetic nervous system (PNS), the autonomic nervous system (ANS) plays a pivotal role in connecting physiological signals to anxiety. The ANS comprises the sympathetic nervous system (SNS) and the parasympathetic nervous system (PNS). When an individual experiences anxiety, the typical response involves the activation of the SNS and the inhibition of the PNS[66]. Slow breathing can potentially enhance the activation of the PNS[67]. Both sympathetic and parasympathetic activities can influence the spontaneous sinus node depolarization, resulting in alterations in cardiac rate and rhythm[68]. For instance, HRV metrics, which can be extracted from ECG and PPG signals, provide insights into how the balance between the SNS and PNS influences cardiac activity[69].

Several studies[70,71] reviewed here have indicated that the HRV extracted from the ECG diminishes when the subject feels anxious. Decreased time-domain HRV metrics, such as RMSSD[33,37,38,44,45,47,50], SDNN[35,37,38,44,45,47,50], and pNN50[35,37,38,47,49,50] have consistently been used in studies to detect anxiety, demonstrating their effectiveness in identifying anxiety-induced reductions in parasympathetic activity. These features are similarly robust when extracted from PPG signals, with observed reductions in RMSSD[20,22,50] and SDNN[20,22,50] reported during anxiety states. In addition to HRV, HR itself has proven to be a reliable marker of anxiety, with increases in HR commonly reported during anxiety episodes. This trend is observed in both ECG-derived measures[36,43–47,50] and PPG-derived measures[19,20,22,42]. Frequency-domain features, such as LF power and HF power[34,35,37,38,45], and VLF power[34,37,38], extracted from ECG, along with the LF/HF ratio–an indicator of autonomic shifts reflecting increased sympathetic dominance in anxiety states-extracted from both ECG[37,38,45,47] and PPG[18,22], are crucial for understanding the balance between sympathetic and parasympathetic activity during anxiety. Together, these features provide a detailed understanding of autonomic changes associated with anxiety. The consistent findings across studies of both ECG and PPG signals highlight the reliability of time- and frequency-domain HRV metrics for accurate anxiety detection.

Complementing HRV metrics, EDA provides a direct measure of SNS activity, making it a valuable signal for anxiety detection, as well. Unlike HRV, which reflects the balance between SNS and PNS, EDA specifically captures sympathetic arousal associated with heightened anxiety states. Commonly used features, such as mean EDA[18,20,44–46], peak SCR amplitude[19,20,44,47], and SD EDA[18,20,44,45], capture both baseline arousal and response intensity with high sensitivity to anxiety-induced SNS changes.

ADs can also lead to respiratory changes. Ihmig et al.[46] reported an increase in respiration rates during anxiety. Haritha et al.[41] suggested that breathing patterns could provide ample information for anxiety detection, but they did not specify how these patterns changed with anxiety. Features such as the mean inspiratory to expiratory time ratio[39,40,47], mean and standard deviation of respiratory rate, as well as mean interbreath intervals[39,41,47], which were among the most commonly used respiratory features in this review, could be promising for detecting anxiety states.

The reviewed studies show that multi-modality approaches generally outperform single-modality methods in terms of accuracy (81.94 vs. 76.85%), likely due to the integration of complementary signals. However, fewer studies reported multi-modality results (16 vs. 27 for single-modality), and most combinations were based on only one or a few studies. The findings should be interpreted with caution, as the limited number of studies and variability in methods, such as different sample sizes, anxiety induction and assessment tools, as well as data preprocessing and machine learning methods, preclude conclusive comparisons of signal efficacy. Signals reflecting ANS activity, such as ECG, PPG, and EDA, demonstrated higher accuracies as compared to RSP, which indirectly reflects ANS activity. Interestingly, integrating RSP with ANS-related signals, such as EDA & RSP (88.00%, based on one study[47]) or ECG, EDA & RSP (82.46%, based on two studies[47,48]), achieved pooled accuracies higher than EDA alone, suggesting that RSP may enhance detection performance. However, RSP results were based on four studies only and exhibited the widest accuracy range (53.00–97.90%), indicating significant methodological differences and limited representation. Among single modalities, ECG yielded the most reliable results, supported by the highest number of studies and a pooled accuracy of 80.42%. Furthermore, many of the highest-accuracy results stem from basic early-fusion approaches (e.g., simple feature concatenation) with minimal feature selection or validation[21,22,44–46,49,50]. In contrast, studies employing structured feature selection or decision-level fusion (e.g.,[42,43,47,48])

### a) Limitation levels across studies

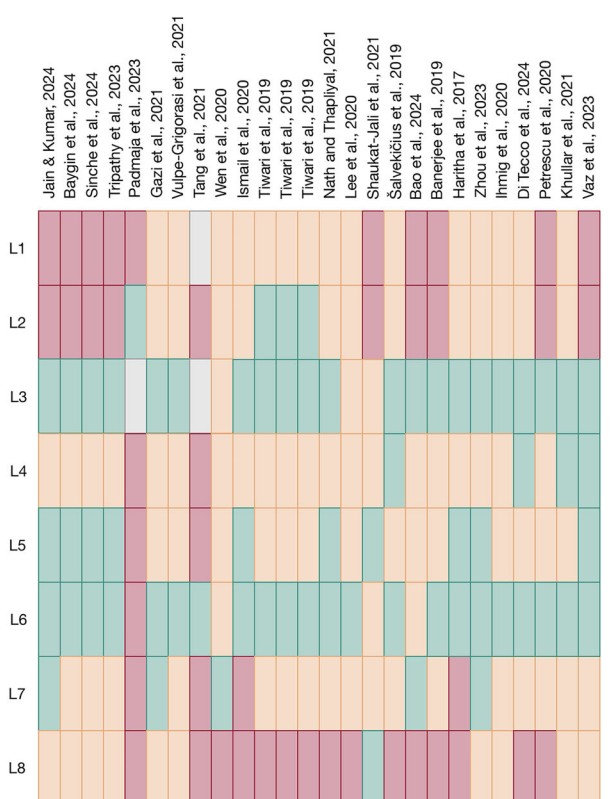

### b) Distribution of limitation levels by category

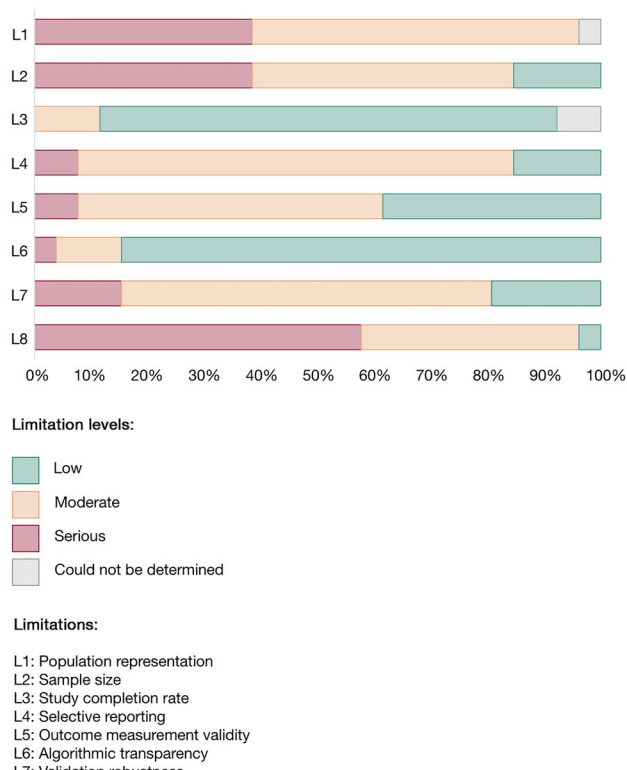

**Fig. 7 | Limitation levels across included studies and their distribution by category. a** Limitation levels across studies: Heatmap summarizing the limitation levels for population representation, sample size, study completion rate, selective reporting, outcome measurement validity, algorithmic transparency, validation robustness, and reproducibility (L1–L8) for each study. Levels are categorized as Low (green), Moderate (orange), Serious (red), or Could Not Be Determined (gray). **b** Distribution of limitation levels by category: Bar chart illustrating the proportion of studies rated as Low, Moderate, Serious, or Could Not Be Determined for each limitation category (L1–L8), highlighting the prevalence of Serious limitations in population representation (L1) and sample size (L2). **Note:** L Limitation.

reported more moderate yet likely generalizable results. These studies applied methods such as feature permutation importance[42,47], chi-squared tests[42], sequential feature selection (SFS)[42,46], and regression-based ranking to refine models and mitigate redundancy. This pattern suggests that some of the highest-reported accuracies may be inflated by overfitting rather than true multimodal complementarity, emphasizing the need for rigorous feature selection. Given the limited evidence for both multi- and single-modality approaches, further research with standardized protocols and larger datasets is essential. Despite these limitations, leveraging ANS-related signals and integrating diverse modalities remain promising strategies for improving anxiety detection.

We also observed that technology can discern anxiety across various stimuli and experimental paradigms, including public speaking tasks[19–21,35,44,45,49], virtual reality-based exposure to phobias such as spiders and heights[33,43,46–48], video-based anxiety induction using general anxiety-inducing clips, horror trailers, and driving scenarios[18,30,31,42], workplace stress monitoring[37–39], and tasks involving relaxation, gaming, or social anxiety[22,36,40]. These findings highlight the adaptability of physiological signal-based methods to a wide range of anxiety-inducing challenges, supporting their potential utility in diverse real-world scenarios.

While we cannot advocate for ECG-, EDA-, PPG- and RSP-based wearable technologies as the sole method for detecting anxiety, their value and potential utility in clinical practice as a supportive tool is clear and only expected to grow along with the advancement and enhanced robustness of these technologies. Unlike traditional 12-lead ECGs, which are typically used in clinical settings and require professional administration, WDs can be used continuously in naturalistic environments. This makes them

especially valuable as a preliminary screening tools or triage mechanisms for patients in rural areas or developing countries who face significant access barriers to prompt medical and mental health care. By providing an early indication of potential anxiety, they can facilitate timely referrals and access to necessary care[72,73]. Additionally, wearable technologies are beneficial for individuals with Autism Spectrum Disorder or others who face challenges with communication and self-reflection[74]. Such individuals may find traditional self-reporting scales difficult, making WDs an attractive alternative for objectively detecting anxiety and ensuring more accessible and timely treatment. Although current research is still exploring the capability of these technologies to distinguish anxiety from other mental health disorders with overlapping symptoms, initial findings on PPG have identified distinctions in HRV between patients with anxiety and depression[75,76]. Nevertheless, further multidisciplinary investigation is essential to determine the specific sensitivity and accuracy of wearables in this context and identify discriminating features. For clinicians in a rapidly emerging precision medicine world, these technologies are poised to be integrated in traditional diagnostic and monitoring protocols of patients diagnosed with social phobia, psychological stress, or anxiety. Wearable technologies provide a dynamic view of patients' progress, enabling for real-time therapeutic adjustments and effectiveness assessments, complementing traditional symptom evaluations during in-person consultations. Importantly, considering the high prevalence of AD in adolescents throughout the world, providing technology-driven tools for assessment, monitoring and treatment is invaluable for a generation that is both technology reliant and tech-savvy.

Early mHealth research on mental health primarily focused on text messaging interventions for psychological support. Studies have shown that

text messaging, or short message service (SMS), is effective for addressing mental health issues, including anxiety[77]. Moreover, research in telemental health has been centered around creating accessible virtual mental health consultations for patients. Such studies indicate that telemental health effectively addresses mental health challenges and enhances patient accessibility[78]. Given the advancements in wearable technologies, there's potential for a more integrated digital healthcare approach. Specifically, wearable signals could be incorporated into a wide array of digital mental health tools. This technology might prove valuable in tandem with telemental health consultations, as it offers objective physiological data in addition to subjective self-reported symptoms of anxiety or social phobias. This could provide clinicians with a more comprehensive, informed and data-driven clinical overview. However, more research is needed to validate this technology before endorsing it as an additional tool and rigorous regulatory measures should be applied towards identifying and approving legitimate interventions.

This review highlights significant advancements in the field of physiological signal-based anxiety detection but is not without limitations. The scarcity of available studies - most multi-modality results were based on single studies, and all single-signal modalities except ECG had a total of six or fewer studies - limits the generalizability of our findings. Results from both single- and multi-modality studies showed a wide range of accuracies, emphasizing study-specific variability. For instance, Zhou et al. reported accuracies of 64.50 and 99.00% for ECG and EDA signals when tested on the APD and WESAD datasets, respectively, highlighting the influence of factors such as sample size, anxiety induction methods, and measurement tools on model performance. Many of the reviewed studies relied on small sample sizes and lacked standardization in anxiety induction methods, feature extraction, and validation approaches. These inconsistencies make it challenging to draw definitive conclusions about the reliability of these technologies across diverse settings. While the reviewed studies focused on controlled anxiety induction in healthy participants, validating wearables for anxiety screening requires both cross-sectional studies comparing diagnosed anxiety patients to healthy controls using standardized tools like Generalized Anxiety Disorder Assessment scale[79], and longitudinal studies assessing whether wearables can detect emerging anxiety disorders. Without these validations, their ability to differentiate temporary stress from chronic anxiety remains uncertain. Gold-standard randomized controlled trials remain essential for establishing clinical validity[80]. Further research should also explore their efficacy in detecting general phobias, psychological stress, and specific anxiety triggers. Moreover, it is crucial to acknowledge that certain medical conditions, such as cardiovascular diseases, can influence ECG and PPG readings. This confounding factor might compromise the precision of anxiety detection through this method[81]. Hence, when executing ECG and PPG analysis for anxiety detection, researchers should tread with caution. Participants with medical issues that might skew the accuracy of ECG and PPG signals should either be omitted from the study, or their conditions should be factored into the data analysis.

With a growing number of people affected by ADs all over the world, there is a pressing need for a dependable anxiety detection system suitable for home use. Gazi et al.[47] pointed out the potential challenges in generalizing the model to real-world settings, which often entail multiple stimuli differing from controlled laboratory conditions. Individuals diagnosed with specific mental disorders might react differently to identical stimuli compared to their healthy counterparts. Expert clinicians have highlighted that while healthy individuals typically exhibit moderate breath-to-breath variability, those with ADs tend to show reduced variability in the same parameter[82,83]. Some research[84,85] has shown that the HRV of individuals with mental disorders (e.g., depression and anxiety) is lower compared to the general healthy populace. However, only one study[41] encompassed data from both healthy participants and those diagnosed with ADs. Based on our review, several recommendations can guide the development of more reliable, technology-driven AD assessment. Future studies should collect biosignals from larger and more diverse populations, with detailed information on age, gender, health status, and ethnicity. Data should capture both calm and anxious states from the same individuals, across a range of mental health states, and ideally in real-world rather than only controlled laboratory settings. Synchronized multimodal signals such as ECG, EDA, PPG, and RSP should be prioritized to better capture the multifactorial markers of anxiety. Additionally, usability issues should be addressed, prioritizing smaller, user-friendly designs that minimize distraction during daily activities. Finally, advanced classifiers should be leveraged to move beyond simple two-level detection, and models should be evaluated using a broader set of metrics, including MCC, ACC, F1-score, sensitivity, and specificity, to enable comprehensive and comparable performance assessments across studies.

The suggestions put forth can prove instrumental in creating reliable and varied datasets. Such datasets can enable researchers to train models with enhanced accuracy for anxiety detection. By sharing these datasets and generalizing their findings across different populations, the robustness and accuracy of machine learning models in real-world scenarios can be improved. Ultimately, these datasets may pave the way for more efficient mental health assessments. The advent of smaller WDs allows for continuous monitoring of bio-signals during an individual's daily activities, promoting personalized mental health detection and timely interventions. Leveraging advanced machine learning algorithms holds promise in refining the accuracy of anxiety detection. Adopting a consistent evaluation metric would simplify comparisons across various studies and methodologies in the realm of anxiety detection.

## Conclusions

This study aimed to assess the efficacy of ECG, EDA, PPG and RSP signals, gathered through various WDs, in the detection of anxiety, while also summarizing the types of WDs used and the features extracted from the different signals. One constraint of our study is that based on the inclusion/exclusion criteria, it only encompassed 26 studies, potentially limiting the broader applicability of our findings. ECG emerged as the most reliable single modality, with robust performance across studies, underscoring its reliability in monitoring psychophysiological changes associated with anxiety states. RSP showed the widest variability and lowest pooled accuracy, underscoring the need for methodological refinement. Although multimodal approaches generally outperformed single-modality methods, the limited number of reviewed studies indicates the need for further research. Most reviewed studies used small datasets, focused on healthy participants, and lacked standardized protocols, limiting the generalizability of the results. Addressing these issues, such as variability in anxiety induction methods, sample sizes, and evaluation metrics, is crucial for achieving comparable outcomes. Considering that episodes of anxiety can manifest unpredictably, it is crucial for individuals to possess a portable WD that offers real-time anxiety evaluation. However, a majority of the studies we reviewed relied on pre-existing databases and were restricted to retrospective signal analysis. Early identification of functional impairments during daily tasks is paramount for prompt clinical intervention. Wearable technologies have immense potential as complementary tools in mental health care, facilitating continuous monitoring, early detection, and timely intervention. By addressing current limitations and focusing on robust datasets and standardized methodologies, research can advance the development of reliable, real-time, technology-driven anxiety detection systems.

## Data availability

This study is a systematic review and did not generate any new datasets. All data supporting the findings are derived from previously published studies, which are cited in the article and detailed in Supplementary Data 1 and Supplementary Data 2.

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

## Author contributions

M.E., K.M. and H.L. conducted the literature searches and analyzed the results. A.K., H.F.J., K.K., M.D.V., D.G. and C.M. oversaw the entire review process, including manuscript review. All authors reviewed the manuscript.

## Funding

## Competing interests

The authors declare no competing interests. M.E. serves as an Associate Editor for *npj Biosensing* and had no involvement in the review or editorial handling of this manuscript.

## Additional information

**Peer review information** : *Communications Medicine* thanks Raymundo Cassani and the other, anonymous, reviewer(s) for their contribution to the peer review of this work.

