## [Transparent Peer Review file · Communications Medicine]

Wearable Devices for Anxiety Assessment: a Systematic Review

Corresponding Author: Professor Mohamed Elgendi

Version 0:

Reviewer comments:

Reviewer #1

(Remarks to the Author)

I found this to be a very comprehensive and well-done review. The authors covered every aspect of the existing literature exploring anxiety by inducing it in various ways. The results are encouraging but have not been applied to clinical samples (which the authors recognize. Hopefully the review will spur on more research that includes wearables with anxiety patients.

Reviewer #2

(Remarks to the Author)

In this study, the authors presented a comprehensive review of anxiety assessment based on wearable devices published within the last year. The manuscript effectively summarizes recent research trends involving physiological signals from wearable devices, including ECG, PPG, and respiration. The review article is likely to be highly beneficial for researchers from various fields.

However, I have a minor question regarding the authors' decision to limit the research period to just one year (2024–2025). Other analyses and perspectives presented in the paper are well-performed in my opinion.

Reviewer #3

(Remarks to the Author)

Major comments

Introduction

1. Since HR and metrics derived from it can be computed from the ECG and PPG signal, it would be a good idea to have a discussion on the advantages/disadvantages of one vs the other (e.g. need to two hands for ECG). Pointing to relevant studies that compare those metrics in the context of wearable devices.

Methods > Pooled accuracy calculation

2. Since anxiety measurement is reported in different ways in the studies, it is important to mention if the accuracy was computed always binary classification, and how this binarization was done in each study.

Minor comments

Introduction

3. Term "Patients" may be replaced by "People" or "Participants", as they are not always diagnosed.

4. Typo "AD1"

5. "Systematic assessments of these biosignals for anxiety detection are limited". Please explain what are these limitations in other reviews, citing those other reviews and how this review overcomes those limitations.

Methods > Data extraction

6. Mention that an issue with ACC is that it can be biased if the number of participants per classes are not balanced. And how other metrics such as F-score aim to address that.

Method > Limitation assessment criteria

7. Description of L8 is duplicated.

Results

8. Consider to replace Subfigure (1b) in Figure2 using a Venn's four-set diagram using ellipses, for easier visualization of the reviewed and the modalities used in them.

9. Some studies, e.g. Jain&Kumar2024, use single modality (ECG and EDA) as well as combined (ECG+EDA). This study should be counted in the three categories: ECG, EDA and ECG+EDA. As it is done in Table1 and Table2. This for Subfigure (1b) in Figure2 and the text in the "Results > Study selection" section.

10. Skin conductance response is mentioned in the Table 1, but not mentioned in the "Introduction" nor "Methods" sections. It should be explained in there. Same goes for skin temperature.

Results > Wearable devices employed across studies

11. "The ECG signals typically indicate the timing of heartbeats and provide insights into HR and rhythm." Is not the same HR and rhythm?

12. This section presents descriptions of RSP, EDA and PPG. This information should be in the "Introduction" rather than in the "Results" section.

Version 1:

Reviewer comments:

Reviewer #3

(Remarks to the Author)

Thank you for addressing my concerns in the revised manuscript.

The manuscript is a well-done comprehensive review of anxiety assessment, which will be useful to provide an quick overview on the field.

Reviewer 1

I found this to be a very comprehensive and well-done review. The authors covered every aspect of the existing literature exploring anxiety by inducing it in various ways. The results are encouraging but have not been applied to clinical samples (which the authors recognize). Hopefully the review will spur on more research that includes wearables with anxiety patients.

Author Response

We sincerely thank the reviewer for the positive and encouraging feedback. We are glad that the comprehensiveness and structure of the review were well-received. As noted, while most studies to date focus on induced anxiety in controlled settings, we agree that future work should explore applications in clinical populations. We have emphasized this research gap in the discussion section to encourage further exploration and validation in real-world clinical contexts.

Author Action

No action required.

Reviewer 2

In this study, the authors presented a comprehensive review of anxiety assessment based on wearable devices published within the last year. The manuscript effectively summarizes recent research trends involving physiological signals from wearable devices, including ECG, PPG, and respiration. The review article is likely to be highly beneficial for researchers from various fields.

- 1. However, I have a minor question regarding the authors' decision to limit the research period to just one year (2024–2025). Other analyses and perspectives presented in the paper are well-performed in my opinion.**

Author Response

We thank the reviewer for this careful reading. We would like to clarify that our review includes studies published between **01 January 2014 and 01 January 2025**, as stated in the “Inclusion and Exclusion Criteria” section of the Methods. There was no restriction to a one-year period. We have reviewed the manuscript and found no inconsistencies in the stated timeframe.

Author Action

No changes were made, as the review period was correctly described in the manuscript.

Reviewer 3

Major comments

Introduction

1. Since HR and metrics derived from it can be computed from the ECG and PPG signal, it would be a good idea to have a discussion on the advantages/disadvantages of one vs the other (e.g. need to two hands for ECG). Pointing to relevant studies that compare those metrics in the context of wearable devices.

Author Response

We thank the reviewer for the insightful suggestion. In response, we have added a comparative discussion at the end of the Introduction, outlining the strengths and limitations of ECG- vs. PPG-derived heart rate metrics for wearable anxiety detection. This includes considerations of sensor placement, signal fidelity, motion sensitivity, and overall usability. We have also cited studies that compare HRV extraction from both modalities in wearable settings.

Author Action

Revised the last paragraph of the introduction, it now reads as:

- *“ECG and PPG features, particularly heart rate variability (HRV), are widely used to detect anxiety disorders (ADs) \cite{Lyzwinski2023}. ECG signals indicate the timing and rhythm of heartbeats by capturing the heart’s electrical activity with high fidelity, enabling accurate derivation of HR and HRV metrics. However, ECG typically requires multi-electrode setups (e.g., chest or limb placement), which can limit wearability. In contrast, PPG captures peripheral blood volume changes using optical sensors, commonly worn at the wrist. Light is emitted into the skin, and the amount of light absorbed or reflected varies with blood flow during the cardiac cycle. PPG is widely used to derive cardiovascular parameters such as HR and oxygen saturation. While it offer better wearability, it is also more prone to motion artifacts and signal degradation under poor perfusion. Comparative studies show that PPG- and ECG-derived HRV metrics correlate well under ideal conditions, though PPG may be less reliable in ambulatory or real-world settings \cite{lu2009comparison, lu2009limitations}.”*

Methods > Pooled accuracy calculation

2. Since anxiety measurement is reported in different ways in the studies, it is important to mention if the accuracy was computed always binary classification, and how this binarization was done in each study.

Author Response

We thank the reviewer for this important observation. To address the variability in how classification labels were derived across studies, we added a new column titled *Anxiety Labeling Approach* in the summary tables to specify whether binary or multi-class classification was used and how labels were determined. Additionally, we included a dedicated paragraph in the *Anxiety Measurement Methods Used* section that explains the classification schemes and highlights cases where labels were derived from experimental context rather than the measurement tool itself.

Author Action

Added a column “Anxiety Labelling Approach” in Tables 1 and 2, mentioning whether the labeling was binary or multi-class and how the labels were derived.

Expanded the *Anxiety Measurement Methods Used* section to describe labeling schemes and provide examples of studies where labeling was based on experimental design rather than scale thresholds. The section reads as:

- “*In terms of classification schemes, anxiety was labeled as binary (e.g., anxious vs. non-anxious) in 18 studies (Sinche2024, padmaja, gazi2021respiratory, vulpe2021neural, tang, wen2018toward, Tiwari2019anxiety, tiwari2019stress, nath2021, lee2020, shaukatjali2021, tiwari2019rsp, banerjee2019reckoning, haritha2017automating, zhou2023, ihmig2020line, Khullar21, vaz2023), while four studies used 3-class schemes (e.g., low/moderate/high) (jainkumar2024, petrescu2020, ditecco2024, bao2024), and three studies used 4-class schemes (e.g., normal/mild/moderate/severe) (baygin2024, tripathy2023, salkevicus2019). One study did not report its labeling method (ismail2020early). Most studies that used standardized scales also used them to define class labels. However, in some cases, labels were derived from experimental context. For example, Vulpe-Grigorasi et al. (vulpe2021neural), Gazi et al. (gazi2021respiratory), and Khullar et al. (khullar2021) assigned labels based on exposure to spider-related VR clips versus rest, despite collecting self-reported anxiety levels. Sinche et al. (sinche2024) used the Cognitive Test Anxiety Scale for assessment, but labeled data based on timing, pre-test as anxious and post-test as non-anxious. Banerjee et al. (banerjee2019reckoning) used SAM ratings, but labeled gameplay windows as anxious or non-anxious depending on whether participants played a frustration-inducing or standard Pacman game. Finally, Shaukat-Jali et al. (shaukatjali2021) employed LSAS-SR and SPSQ for anxiety assessment, but classification into anxious versus baseline was based on the timing of data collection relative to the impromptu speech task.”*

Minor comments

Introduction

3. Term "Patients" may be replace by "People" or "Participants", as they are not always diagnosed.

Author Response

We thank the reviewer for this suggestion. We agree that the term "patients" may not always be appropriate, as not all individuals included in the reviewed studies were clinically diagnosed. We have updated the text accordingly to use more neutral terms such as "people" or "participants" where applicable.

Author Action

Replaced "patients" with "people" or "participants" where relevant. For example:

- *“People suffering from ADs typically endure intense levels of fear or distress disproportionate to real events. ADs affect both adults and children. The anxiety/fear occupying the daily lives of these people affects their behaviors, thoughts, and physical health, often hindering their ability to function normally.”*

4. Typo "AD1"

Author Response

Thank you for pointing this out. The typo has been corrected.

Author Action

Corrected "AD1" to "ADs" in the sentence:

- *“However, only 1 in 4 patients are reported to receive treatment for ADs due to lack of awareness, lack of competent or accessible mental health services, as well as lack of trained health care providers\cite{world2017depression}.”*

5. "Systematic assessments of these biosignals for anxiety detection are limited". Please explain what are these limitations in other reviews, citing those other reviews and how this review overcomes those limitations.

Author Response

Thank you for the suggestion. We have now clarified the specific limitations of prior reviews and explained how our review addresses these gaps. We also cited the following key prior works: "Smart Devices and Wearable Technologies to Detect and Monitor Mental Health Conditions and Stress: A Systematic Review" by Hickey et al., "A Systematic Review on Physiology-based Anxiety Detection using Machine Learning" by Shikha et al., "Machine learning in biosignals processing for mental health: A narrative review" by Sajno et al., "Wearable devices for anxiety & depression: A scoping review" by Ahmed et al.

Author Action

We updated the Introduction to cite these previous reviews, describe their limitations, and explain how our review addresses them:

- *“However, systematic assessments of these biosignals for anxiety detection are limited. Prior reviews\cite{hickey2021smart, shikha2025systematic, Sajno20222, Ahmed2023} often focus more broadly on mental health rather than anxiety specifically, lack pooled performance comparisons across modalities or do not systematically assess study limitations. This review examines 26 studies utilizing ECG, RSP, EDA, and PPG signals and their combinations, comparing single- and multimodal approaches. We analyze pooled accuracy, highlight the most frequently studied features and classifiers, and provide insights into the clinical potential of biosignal-based WDs for anxiety detection. This work aims to advance the understanding of WD applications in mental health and inform future research directions.”*

Methods > Data extraction

6. Mention than an issue with ACC is that it can be biased if the number of participants per classes are not balanced. And how other metrics such as F-score aim to address that.

Author Response

Thank you for the suggestion. We have added a clarification on the limitations of using accuracy (ACC) as the primary metric, particularly in the context of class imbalance, and explained how alternative metrics like F1-score address this issue.

Author Action

We updated the text to note:

- *“While all reported metrics (e.g., precision, recall, F1-score) were extracted, accuracy (ACC) was used as the primary metric to ensure consistency, due to being the most consistently reported. However, it is important to note that ACC can be biased when class distributions are imbalanced, as it does not account for false positives or false negatives in the minority class. Metrics like F1-score or Matthews correlation coefficient (MCC) better address this by incorporating both precision and recall, providing a more balanced view of model performance under class imbalance.”*

Method > Limitation assessment criteria

7. Description of L8 is duplicated.

Author Response

Thank you for pointing this out. We have removed the duplicated sentence to avoid redundancy.

Author Action

Updated the text to:

- *“Finally, reproducibility (L8) was assessed based on the availability of datasets, code, and tools for independent replication.”*

Results

8. Consider to replace Subfigure (1b) in Figure2 using a Venn's four-set diagram using ellipses, for easier visualization of the reviewed and the modalities used in them.

Author Response

We appreciate the suggestion. However, creating a Venn diagram would require labeling all 26 included studies and distinguishing studies that used varying signal combinations. This made the visualization overly complex and less readable.

Author Action

No changes made to Figure 2, as the current representation was deemed clearer for summarizing modality usage across studies.

9. Some studies, e.g. Jain&Kumar2024, use single modality (ECG and EDA) as well as combined (ECG+EDA). This study should be counted in the three categories: ECG, EDA and ECG+EDA. As it is done in Table1 and Table2. This for Subfigure (1b) in Figure2 and the text in the "Results > Study selection" section.

Author Response

Thank you for the comment. Such studies were already counted in all relevant categories (e.g., ECG, EDA, and ECG+EDA), as shown in Tables 1 and 2. To improve clarity of the description in the figure title, we now explicitly mention this in the text and added Jain & Kumar (2024) as an illustrative example.

Author Action

Updated the figure title description, it now reads as:

- *“Step 1: Search and selection of studies: Subfigure 1a depicts the wearable signal modalities analyzed: ECG, RSP, EDA, and PPG. Subfigure 1b shows the number of reviewed studies that used each signal modality, including both single- and multi-modality approaches. Some studies contributed to multiple categories; for example, Jain & Kumar\cite{jainkumar2024} reported results using ECG alone, EDA alone, and ECG+EDA combined. These were counted in all relevant categories.”*

10. Skin conductance response is mentioned in the Table 1, but not mentioned in the "Introduction" nor "Methods" sections. It should be explained in there. Same goes for skin temperature.

Author Response

Thank you for pointing this out. The main focus of our review was on four primary signals: ECG, RSP, EDA, and PPG. However, in line with transparency, we included studies employing multimodal approaches that also used additional signals such as skin temperature (SKT) alongside the core modalities. To address this, we have now clarified the role of SKT and similar additional signals in both the *Introduction* and *Methods* sections.

Author Action

In the introduction, we added:

- “EDA, which measures sweat gland activity through skin conductance, is particularly sensitive to emotional arousal and has been correlated with anxiety levels \cite{lee2020, nath2021}. *Some studies\cite{salkevicius2019, shaukatjali2021, bao2024} also included skin temperature (SKT) measurements alongside EDA or PPG, as SKT may capture thermoregulatory or stress-related changes relevant to anxiety detection.*”

In the Methods – Search Strategy and Study Eligibility, we added:

- “*While the search focused on ECG, RSP, EDA, and PPG, studies employing multimodal approaches that included additional signals (e.g. SKT or electromyography) alongside these core modalities were also included.*”

Results > Wearable devices employed across studies

11. "The ECG signals typically indicate the timing of heartbeats and provide insights into HR and rhythm." Is not the same HR and rhythm?

Author Response

Thank you for the comment. While heart rate (HR) and heart rhythm are related, they are not the same. HR refers to the number of heartbeats per minute, while rhythm refers to the temporal pattern and regularity of those beats. Both can offer distinct insights into autonomic nervous system activity, which is relevant in anxiety detection. Therefore, we believe the current phrasing accurately reflects this distinction.

Author Action

No change was made to the sentence.

12. This section presents descriptions of RSP, EDA and PPG. This information should be in the "Introduction" rather than in the "Results" section.

Author Response

Thank you, we fully agree with this suggestion. The descriptions of RSP, EDA, and PPG were moved from the Results section to the Introduction, where they provide better context for the signal modalities analyzed in the review. We also reorganized the "Wearable devices employed across studies" section to focus on the devices themselves, grouped first by those measuring single modalities and then by those capturing multiple modalities. This improves readability and maintains consistency across sections.

Author Action

The descriptions of RSP, EDA and PPG were moved to the introduction, the Introduction section now reads as:

- *“ECG and PPG features, particularly heart rate variability (HRV), are widely used to detect anxiety disorders (ADs) \cite{Lyzwinski2023}. ECG signals indicate the timing and rhythm of heartbeats by capturing the heart’s electrical activity with high fidelity, enabling accurate derivation of HR and HRV metrics. However, ECG typically requires multi-electrode setups (e.g., chest or limb placement), which can limit wearability. In contrast, PPG captures peripheral blood volume changes using optical sensors, commonly worn at the wrist. Light is emitted into the skin, and the amount of light absorbed or reflected varies with blood flow during the cardiac cycle. PPG is widely used to derive cardiovascular parameters such as HR and oxygen saturation. While it offer better wearability, it is also more prone to motion artifacts and signal degradation under poor perfusion. Comparative studies show that PPG- and ECG-derived HRV metrics correlate well under ideal conditions, though PPG may be less reliable in ambulatory or real-world settings \cite{lu2009comparison, lu2009limitations}. Despite decades of research on ECG-derived HRV, its utility in anxiety detection remains debated \cite{elgendi2019assessing}. Respiration cycles involve inspiration (air inflow, diaphragm contraction, lung volume increase) and expiration (air outflow, diaphragm relaxation, lung volume decrease), producing measurable thoracoabdominal motion. Respiratory dysregulations, such as breath-to-breath respiratory instability, frequent sighing, are also common characteristics of ADs\cite{giardino2007anxiety}. Breath rate generally increases under anxiety and decreases under relaxation. Several studies indicated that ECG and RSP features were closely associated with anxiety and stress detection \cite{elgendi2019assessing, Moe2020ML}. EDA reflects changes in skin conductance resulting from sweat gland activity, which is modulated by the autonomic nervous system. These changes occur in response to emotional arousal, stress, or cognitive load, making EDA a valuable marker for anxiety detection \cite{lee2020, nath2021}. Some studies \cite{salkevicius2019, shaukatjali2021, bao2024} also incorporate skin temperature (SKT) alongside EDA or PPG, as it may capture stress-related peripheral vasoconstriction. ”*

The “Wearable devices employed across studies” section in the results was reorganised and now reads as:

- *“The reviewed studies included here used various WDs with diverse configurations and placements to capture ECG, EDA, PPG, and RSP signals for anxiety detection. The sketches of the devices used are provided in Figure \ref{fig:devices}, and the devices that*

each study used are listed in Tables \ref{tab1} and \ref{tab2}. Below, we group the devices based on the signal modalities they recorded in the included studies.

Nine devices were employed to record single signal modalities in the studies reviewed. Out of these, five devices were used to record ECG signals: (1) the SS2LB ECG module \cite{baygin2024, tripathy}; (2) AD8232 sensor \cite{sinche2024}; (3) the Biopac MP-45 system \cite{padmaja}; (4) the MP150 recorder \cite{wen2018toward}; (5) the Zephyr BioHarness 3.0 \cite{zhou2023}. The SS2LB ECG module, used in the Elgendi et al. \cite{elgendi2022dataset} dataset utilized by Baygin et al. \cite{baygin2024} and Tripathy et al. \cite{tripathy} employed a Lead I configuration (right arm (RA) to left arm (LA), left leg (LL) as ground), designed for high-resolution recordings compatible with Biopac systems. The AD8232 sensor, used by Sinche et al. \cite{sinche2024}, consists of a compact analog front-end module placed on the left side of the chest with three electrodes configured in a standard Lead II setup. The Biopac MP-45 system, used by Padmaja et al. \cite{padmaja}, featured an embedded CPU for data acquisition and supported various electrode types, though specific placements were not detailed. The MP150 multi-channel recorder, used by Wen et al. \cite{wen2018toward}, collected ECG signals with electrodes placed on both wrists and the right ankle, sampling at 400 Hz, alongside skin conductance data. Lastly, the Zephyr BioHarness 3.0, used in the APD \cite{apd} dataset utilized by Zhou et al. \cite{zhou2023}, recorded chest-based ECG data with two disposable adhesive electrodes aligned to the breastbone, sampling at 250 Hz. Most of these devices, as illustrated in Figure \ref{fig:devices}a, are attached to the subject with a strap. Two devices were employed to record RSP data: (1) Embletta \cite{haritha2017automating} and (2) an undefined respiration belt \cite{banerjee2019reckoning}. The Embletta device (Figure \ref{fig:devices}c), used by Haritha et al. \cite{haritha2017automating}, contains two elasticity belts with sinusoid conducting coils for thoracic-abdominal movement detection. One belt is placed on the chest, and the other is placed on the subject's abdomen, such that variations of the thoracic and abdominal volumes cause changes in the surface of the section and stretching of elasticity belts. This leads to variation of the self-inductance of the coils and the frequency of their oscillation, hence inducing a current \cite{Gastinger}. Banerjee et al. \cite{banerjee2019reckoning} obtained the RSP using a respiration belt that measures the expansion of the abdomen \cite{1457203} related to the quantity of inspired and expired air. One sensor was used to measure EDA signal, the Grove-GSR Sensor V1.2, utilized by Zhou et al. \cite{zhou2023}. It measured EDA signals at a sampling rate of 50 Hz using electrodes placed on the index and ring fingers, with the device enclosed in a 3D-printed module secured on both wrists via Velcro straps. One sensor, the MAX30100 sensor, employed by Sinche et al. \cite{sinche2024}, recorded PPG signals from the students' index finger.

Five devices were employed to record multiple signal modalities in the studies reviewed: (1) the RespiBAN device \cite{jainkumar2024, zhou2023, vaz2023} for recording ECG and EDA signals; (2) the OMSignal smartshirt \cite{Tiwari2019anxiety, tiwari2019stress, tiwari2019rsp} for ECG and RSP signals; (3) BiTalino \cite{gazi2021respiratory, Khullar21, ihmig2020line, vulpe2021neural} for ECG, EDA and RSP signals; (4) Empatica E4 \cite{vaz2023, jainkumar2024, zhou2023, shaukatjali2021, salkevicius2019, lee2020}

bao2024} for EDA and PPG signals; (5) Shimmer3 GSR+ \cite{ditecco2024, petrescu2020} also for EDA and PPG signals. The RespiBAN device, a chest-worn sensor capable of capturing ECG, EDA, EMG, RSP, body temperature, and acceleration, was used in the WESAD dataset \cite{wesad} and employed by Jain and Kumar \cite{jainkumar2024}, Zhou et al. \cite{zhou2023}, and Vaz et al. \cite{vaz2023}. The OMSignal smartshirt (Figure \ref{fig:devices}b) used by Tiwari et al. \cite{Tiwari2019anxiety, tiwari2019stress, tiwari2019rsp} contains a close-fitting T-shirt and an OMBox. The T-shirt has sensors that are directly knit into the textile. It leverages conductive yarns with electromechanical properties. The OMBox is fastened to the T-shirt using snap buttons. A battery that lasts for 30 hours of real-time usage powers the OMBox. The built-in Bluetooth module connects to the mobile phone and transfers data. As shown in Figure \ref{fig:devices}d, the BITalino respiration belt, which was used in the dataset by Ihmig et al., 2020 \cite{ihmig2020electrocardiogram} and used in four studies \cite{gazi2021respiratory, Khullar21, ihmig2020line, vulpe2021neural} is strapped around the subject's chest for recording changes in thoracic movement during breathing. The Empatica E4 wristband (Figure \ref{fig:devices}e), used in the WESAD dataset \cite{wesad} utilized in three studies \cite{vaz2023, jainkumar2024, zhou2023} and four further studies \cite{shaukatjali2021, salkevicius2019, lee2020, bao2024}, is worn on the non-dominant wrist and records EDA at 4 Hz, PPG at 64 Hz, body temperature at 4 Hz, and three-axis acceleration at 32 Hz. The Shimmer3 GSR+ device (Figure \ref{fig:devices}f), used in two reviewed studies \cite{ditecco2024, petrescu2020}, measures EDA using electrodes placed on the index and middle fingers, capturing PPG signals via an optical pulse probe on the ring finger and transmitting data in real-time via Bluetooth.”